# TS-MOF: Two-Stage Multi-Objective Fine-tuning for Long-Tailed Recognition

**Zhe Zhao**[1,2]   **Zhiheng Gong**[1]   **Pengkun Wang**[1,3]*   **HaiBin Wen**[2]   **Cankun Guo**[5]
**Bo Xue**[2]   **Xi Lin**[2]   **Zhenkun Wang**[4]   **Qingfu Zhang**[2]   **Yang Wang**[1,3,6]*
[1]University of Science and Technology of China   [2]City University of Hong Kong
[3]Suzhou Institute for Advanced Research, USTC   [4]Southern University of Science and Technology
[5]Nanchang University   [6]Anhui Provincial Key Laboratory of High Performance Computing

## Abstract

Long-Tailed Recognition (LTR) presents a significant challenge due to extreme class imbalance, where existing methods often struggle to balance performance across head and tail classes. Directly applying multi-objective optimization (MOO) to leverage multiple LTR strategies can be complex and unstable. To address this, we propose TS-MOF (Two-Stage Multi-Objective Fine-tuning), a novel framework that strategically decouples feature learning from classifier adaptation. After standard pre-training, TS-MOF freezes the feature backbone and focuses on an efficient multi-objective fine-tuning of specialized classifier heads. The core of TS-MOF's second stage lies in two innovations: Refined Performance Level Agreement for adaptive task weighting based on real-time per-class performance, and Robust Deterministic Projective Conflict Gradient for stable gradient conflict resolution and constructive fusion. This approach enables effective synergy between diverse LTR strategies, leading to significant and balanced performance improvements. Extensive experiments on CIFAR100-LT, ImageNet-LT, and iNaturalist 2018 demonstrate that TS-MOF achieves state-of-the-art results, particularly enhancing tail class accuracy (e.g., +3.3% on CIFAR100-LT IR=100 tail) while improving head class performance, all within a remarkably short fine-tuning period of 20 epochs. We provide the detailed code in https://github.com/DataLab-atom/TS-MOF.

## 1   Introduction

The ubiquitous long-tailed distribution of real-world visual data presents a formidable obstacle to training deep learning models that can generalize well across all categories [21, 3]. Models trained on such imbalanced datasets inevitably develop a bias towards data-rich head classes, leading to significantly deficient performance on data-scarce tail classes. Although numerous Long-Tailed Recognition (LTR) strategies have been proposed—spanning resampling [4, 2], loss re-weighting [19, 5], and advanced representation learning [13, 29]—they often suffer from the "seesaw dilemma": efforts to enhance tail class performance frequently come at the cost of head class accuracy, exposing an inherent trade-off that is difficult to perfectly reconcile with a single strategy.

Given the limitations of individual strategies and the tendency for different LTR methods to exhibit unique strengths across various class regions, "multi-strategy collaboration" has emerged as an attractive research direction. Recently, ideas from Multi-Objective Optimization (MOO) [26, 6] and strategy fusion [27] have begun to be introduced into the LTR domain, attempting to find more balanced solutions by jointly optimizing multiple conflicting objectives. However, directly applying multiple complex LTR strategies to end-to-end training of an entire network can lead to

---

*Corresponding author.

39th Conference on Neural Information Processing Systems (NeurIPS 2025).

substantial computational overhead and tuning difficulties. It may also result in suboptimal feature representations and unstable training dynamics due to potential conflicts between strategies and interference with the feature learning process. This complexity has limited the widespread application and usability of advanced MOO concepts in the LTR field.

To address these challenges, we propose TS-MOF (Two-Stage Multi-Objective Fine-tuning), a novel LTR framework. The core idea of TS-MOF lies in strategic decoupling: first, high-quality generic feature representations are obtained through standard pre-training; subsequently, with this feature backbone frozen, the focus shifts to efficiently fine-tuning a set of classifier heads specifically designed for LTR using multi-objective optimization. This two-stage design significantly lowers the barrier to combining multiple LTR strategies with MOO, as the complex MOO process acts only on the classifier heads, which have fewer parameters, and without the risk of corrupting well-learned features. This enhances overall training robustness and methodological ease of use. Notably, this targeted multi-objective adaptation process is highly efficient; we demonstrate that it can be completed within a very short fine-tuning period (e.g., just 20 epochs) while achieving excellent performance.

The efficacy of TS-MOF hinges on its sophisticated multi-objective optimization mechanism in the second stage, driven by two core innovations that ensure effective integration and synergy of knowledge from different LTR strategies:

1. **Refined Performance Level Agreement (R-PLA) Weighting:** R-PLA is an adaptive dynamic task weighting mechanism that surpasses traditional fixed or simple heuristic weight assignments. By evaluating the instantaneous accuracy of each LTR strategy ("task") on various classes in real-time and combining this with its cosine similarity to a preset reference performance pattern (transformed by a cube root and non-negative clipping for enhanced robustness and differentiability), R-PLA intelligently identifies and amplifies signals from strategies that contribute most to improving critical class performance at the current learning stage. This ensures that optimization resources are directed towards the most effective paths within the limited fine-tuning window.

2. **Robust Deterministic Projective Conflict Gradient (RD-PCGrad):** To resolve gradient conflicts and negative transfer issues that may arise when updating parameters with multiple LTR strategies, we developed RD-PCGrad. This technique is a significant enhancement over existing conflict-avoidance gradient methods (like PCGrad). By performing deterministic iterative projections and numerically stable aggregation of task gradients (weighted by R-PLA), it forms a Pareto-improving direction. RD-PCGrad not only guarantees the reproducibility and training stability of the multi-task learning process—crucial for convergence in a short time—but more importantly, it enables the constructive fusion of heterogeneous knowledge from different LTR strategies (e.g., some focusing on class balance, others on hard-sample mining), jointly propelling the model towards a more balanced and superior performance state.

During inference, we employ the Evolving Optimal Strategy Selection (EOSS) mechanism [32] to integrate predictions from the different classifier heads. The core performance improvements and unique advantages of TS-MOF, however, primarily stem from the efficient, stable, and synergistic multi-objective fine-tuning process during training, driven by R-PLA and RD-PCGrad on the frozen feature foundation.

We validate the superior performance of the TS-MOF framework through extensive experiments on several challenging LTR benchmark datasets, including CIFAR100-LT, ImageNet-LT, and iNaturalist 2018. The experimental results demonstrate that TS-MOF not only surpasses existing state-of-the-art (SOTA) methods in overall accuracy across various benchmarks but also achieves significant breakthroughs in crucial tail-class recognition accuracy, while also improving or effectively maintaining head-class performance. For instance, on the most challenging CIFAR100-LT (IR=100) benchmark, TS-MOF achieves **56.8%** overall accuracy and **39.9%** tail accuracy, a significant improvement over the previous SOTA method LOS [27] (53.9% overall, 36.6% tail), with head accuracy also increasing from 70.3% to 79.0%—all within an efficient 20-epoch fine-tuning budget. These results fully substantiate that our strategy of two-stage decoupling combined with advanced MOO components offers a robust, user-friendly, and efficient solution for tackling complex long-tailed recognition problems. We will release our code and pre-trained models to foster further research in the community.

## 2 Related Work

### 2.1 Long-Tailed Recognition

Long-Tailed Recognition (LTR) addresses the challenge of learning from datasets where class distributions are heavily skewed, with a few head classes having abundant samples and many tail classes having very few. Initial efforts focused on **data re-sampling**, either by over-sampling tail classes [4] or under-sampling head classes [2, 9]. However, naive over-sampling can lead to overfitting on tail classes, while under-sampling discards valuable data from head classes.

Another prominent direction is **loss re-weighting**, which aims to assign higher importance to samples from tail classes during training. Techniques like Focal Loss [19] down-weight easy examples, indirectly helping with class imbalance, and have inspired further work such as Focal-SAM which combines focal-style weighting with sharpness-aware minimization [18]. More directly, methods like Class-Balanced Loss [5] re-weight the loss based on the effective number of samples per class. Label-Distribution-Aware Margin (LDAM) loss [3] enforces larger margins for minority classes, and Balanced Softmax [25] adjusts the softmax function to account for class frequencies. Other approaches modify the logits directly to achieve re-balancing, such as through Gaussian-based adjustments [15]. The theoretical underpinnings of re-weighting and logit-adjustment have also been explored to provide a unified understanding of their generalization properties in imbalanced learning [30].

**Representation learning and knowledge transfer** techniques seek to learn more robust and generalizable features, or transfer knowledge from head to tail classes. Methods like BBN [33] use a two-branch network to learn representations and perform cumulative learning. Other approaches focus on feature augmentation for tail classes, disentangling feature norms from directions [20], or fusing features from head to tail classes to improve representations [17]. Some research delves into the intrinsic properties of learned features, unveiling and mitigating biases by analyzing perceptual manifolds [22]. Recently, adapting large pre-trained models using visual prompt tuning has emerged as a promising direction, with methods designed to improve performance on tail classes [16, 11]. Multi-expert models, such as RIDE [29], train diverse experts specialized in different parts of the data distribution.

### 2.2 Two-Stage Long-Tailed Learning

Recognizing the difficulty of simultaneously learning good features and a balanced classifier on long-tailed data, two-stage approaches have gained significant traction. The seminal work by Kang et al.[13] proposed decoupling representation learning from classifier learning. In their framework, features are first learned using instance-balanced sampling, and then the classifier is re-trained or adjusted using class-balanced sampling or other techniques like logit adjustment[24]. This decoupling strategy allows the backbone network to learn general and high-quality features without being overly biased by the imbalanced data distribution during the initial stage. Subsequent works have built upon this paradigm. For example, LOS [27] explores an optimal strategy within a two-stage framework by adaptively combining multiple re-balancing strategies. Our proposed TS-MOF framework also adopts a two-stage design, but uniquely focuses the second stage on a sophisticated multi-objective fine-tuning of specialized classifier heads using novel components like R-PLA and RD-PCGrad.

### 2.3 Multi-Objective Optimization in Deep Learning

Multi-Objective Optimization (MOO) is concerned with simultaneously optimizing multiple, often conflicting, objectives. In deep learning, MOO has been increasingly applied to multi-task learning (MTL) and problems with inherent trade-offs. Sener and Koltun [26] framed MTL as MOO, proposing methods to find Pareto optimal solutions by optimizing a weighted sum of losses or manipulating gradients. Gradient manipulation techniques are particularly relevant. For instance, MGDA (Multiple Gradient Descent Algorithm)[7] finds a common descent direction that improves all tasks. PCGrad (Projective Conflict Gradient)[31] mitigates conflicting gradients between tasks by projecting a task's gradient onto the normal plane of another task's gradient if they conflict. Our RD-PCGrad builds upon such principles to ensure stable and deterministic conflict resolution. The goal of MOO is often to find a set of solutions on the Pareto front, where no objective can be improved without degrading at least one other objective [6]. While traditional MOO often involves complex evolutionary algorithms,

recent adaptations for deep learning focus on gradient-based approaches suitable for network training. Applying MOO to LTR is a nascent but promising direction. For example, Mahapatra and Rajan [23] proposed a framework unifying MTL and LTR, implicitly leveraging MOO principles to balance learning across diverse class groups and tasks. Our TS-MOF explicitly formulates the second-stage fine-tuning as an MOO problem, leveraging dynamic weighting (R-PLA) and conflict-aware gradient modulation (RD-PCGrad) to achieve a balanced performance improvement across head, medium, and tail classes.

## 3  Methodology

To address the challenges of long-tailed image recognition, we propose a novel two-stage training framework. This framework first learns generic features through standard training and then, in the second fine-tuning stage, focuses on introducing advanced Multi-Objective Optimization (MOO) strategies to effectively leverage the complementary strengths of various long-tailed learning techniques. Compared to traditional single-stage or simpler two-stage long-tailed methods, this design allows for more flexible integration and balancing of multiple strategies, thereby achieving more balanced and comprehensive performance improvements across all parts of the long-tailed distribution. In contrast to applying MOO throughout all stages, our method simplifies the optimization process by separating feature learning and classifier adaptation. This ensures the acquisition of high-quality base features and enables the MOO in the second stage to focus more effectively and efficiently on resolving imbalance issues at the classifier level.

### 3.1  Stage 1: Generic Feature Pre-training

Let $\mathcal{D}_{\text{train}} = \{(x_i, y_i)\}_{i=1}^{M_{\text{train}}}$ be the long-tailed training dataset, where $x_i \in \mathcal{X}$ represents the input image, $y_i \in \mathcal{Y} = \{0, \ldots, K-1\}$ is its class label, $M_{\text{train}}$ is the total number of training samples, and $K$ is the total number of classes. We employ a Convolutional Neural Network (CNN) as an encoder $E(\cdot; \theta_E) : \mathcal{X} \to \mathbb{R}^{D_F}$, which maps the input image to a $D_F$-dimensional feature space, with parameters $\theta_E$. This encoder is connected to a classifier head $H_{\text{S1}}(\cdot; \theta_{H_{\text{S1}}}) : \mathbb{R}^{D_F} \to \mathbb{R}^{K}$, with parameters $\theta_{H_{\text{S1}}}$. The objective of this stage is to learn a high-quality generic feature representation $\theta_E^*$ by minimizing the standard cross-entropy loss $\mathcal{L}_{\text{CE}}$:

$$\min_{\theta_E, \theta_{H_{\text{S1}}}} \mathbb{E}_{(x,y) \sim \mathcal{D}_{\text{train}}} \left[ \mathcal{L}_{\text{CE}} \left( H_{\text{S1}} \left( E(x; \theta_E) \right), y \right) \right] \tag{1}$$

Optionally, Label Smoothing (LS) can be applied:

$$\mathcal{L}_{\text{CE\_LS}}(\mathbf{z}, y) = (1 - \alpha)\mathcal{L}_{\text{CE}}(\mathbf{z}, y) + \frac{\alpha}{K} \sum_{j=0}^{K-1} \mathcal{L}_{\text{CE}}(\mathbf{z}, j) \tag{2}$$

where $\mathbf{z} \in \mathbb{R}^{K}$ are the logits output by the model, and $\alpha$ is the smoothing factor. This stage is trained for a predetermined number of epochs $E_{\text{S1}}$, and the encoder parameters $\theta_E^*$ that achieve the best performance on a validation set are saved.

### 3.2  Stage 2: Multi-Objective Long-Tailed Fine-tuning

This stage constitutes the core innovation of our method. It focuses on fine-tuning a set of classifier heads specifically designed for long-tailed distributions using advanced MOO strategies, based on the high-quality and frozen feature encoder $E(\cdot; \theta_E^*)$ obtained in the first stage. This "encoder-frozen, multi-head fine-tuning" strategy decouples the complex long-tailed problem into two phases: generic feature learning and specialized classifier adaptation, significantly reducing optimization difficulty. More importantly, it provides a flexible framework that conveniently allows various existing single-objective long-tailed learning techniques (often manifested as specific loss functions) to be transformed into independent "tasks" within our MOO framework, thereby exploring their synergistic potential.

#### 3.2.1  Multi-Head Architecture and Task Definition

We equip the model with a multi-head output layer. Let $\mathcal{T} = \{T_1, \ldots, T_N\}$ be the set of $N$ selected learning tasks. Each task $T_k \in \mathcal{T}$ corresponds to a specific long-tailed learning strategy and is

associated with a unique loss function $\mathcal{L}_k$ and an independent classifier head $H_k(\cdot; \theta_{H_k}) : \mathbb{R}^{D_F} \to \mathbb{R}^K$. These loss functions can include standard Cross-Entropy ($\mathcal{L}_{CE}$), Balanced Softmax ($\mathcal{L}_{BS}$), Label-Distribution-Aware Margin loss ($\mathcal{L}_{LDAM}$), etc. In this fine-tuning stage, only the set of parameters for all classifier heads $\theta_H = \{\theta_{H_1}, \ldots, \theta_{H_N}\}$ is optimized.

### 3.2.2 Multi-Objective Optimization Framework

When employing multiple tasks ($N > 1$), we face a multi-objective optimization problem, the goal of which is to simultaneously minimize the expected losses of all selected tasks:

$$\min_{\theta_H} \mathbf{L}(\theta_H) = \min_{\theta_H} \left( \mathbb{E}[\mathcal{L}_1(\theta_H)], \ldots, \mathbb{E}[\mathcal{L}_N(\theta_H)] \right) \tag{3}$$

To effectively address this problem, we design and integrate the following key components:

**Dynamic Task Weighting (Refined PLA)**   To adaptively adjust the relative importance of each task loss $\mathcal{L}_k$ in the overall optimization, we employ an improved dynamic task weighting mechanism. In each training batch of every training epoch $e$, for each task $T_j \in \mathcal{T}$ ($j = 1, \ldots, N$), we compute its performance vector $\mathbf{a}_{j,e}^{(\text{batch})} \in \mathbb{R}^C$ on that batch's data (e.g., per-class accuracy, where $C$ is the number of classes). These performance vectors form a list $A_e^{(\text{batch})} = \{\mathbf{a}_{1,e}^{(\text{batch})}, \ldots, \mathbf{a}_{N,e}^{(\text{batch})}\}$. We designate the last performance vector in the list, $\mathbf{a}_{N,e}^{(\text{batch})}$, as the reference performance $\mathbf{a}_{\text{ref},e}^{(\text{batch})}$ for the current batch. For each performance vector $\mathbf{a}_{j,e}^{(\text{batch})}$ in the list, we compute its cosine similarity $s_{j,e}$ with the reference performance vector:

$$s_{j,e} = \frac{\mathbf{a}_{j,e}^{(\text{batch})} \cdot \mathbf{a}_{\text{ref},e}^{(\text{batch})}}{\max \left( \|\mathbf{a}_{j,e}^{(\text{batch})}\|_2 \cdot \|\mathbf{a}_{\text{ref},e}^{(\text{batch})}\|_2, \epsilon_{\text{sim}} \right)} \tag{4}$$

where $\epsilon_{\text{sim}}$ is a small positive constant ensuring numerical stability. Subsequently, the weight $\beta_{j,e}$ for task $T_j$ is obtained by applying a cube root transformation to the similarity and performing non-negative clipping:

$$\beta_{j,e} = \max(0, \sqrt[3]{s_{j,e}}) \tag{5}$$

The cube root transformation here adjusts the sensitivity of the weights to changes in similarity. Cosine similarity, due to its insensitivity to vector magnitude, can more robustly capture similarities in performance distribution patterns among tasks in a long-tailed context, while non-negative clipping ensures the rational interpretability of weights. The weighted loss is $\mathcal{L}'_k = \beta_{k,e}\mathcal{L}_k$.

**Conflict-Aware Gradient Modulation (Robust & Deterministic PCGrad)**   Even after task weighting, the gradients $\mathbf{g}_k = \nabla_{\theta_H} \mathcal{L}'_k \in \mathbb{R}^{D_{\text{params}}}$ (flattened vectors with respect to all trainable parameters $\theta_H$) corresponding to different weighted losses $\mathcal{L}'_k$ may still conflict. To mitigate negative transfer, we employ a robust and deterministic PCGrad strategy. Let $G^{(0)} = \{\mathbf{g}_1, \ldots, \mathbf{g}_N\}$ be the initial set of gradients, and $\mathbf{h}_k \in \{0,1\}^{D_{\text{params}}}$ be the corresponding gradient existence masks. PCGrad progressively mitigates conflicts between gradients through a series of deterministic, pairwise projection operations. For any pair of gradients $\mathbf{g}'_i$ (current state of gradient $i$) and $\mathbf{g}'_j$, if they conflict ($\mathbf{g}'_i \cdot \mathbf{g}'_j < 0$) and both contribute to the parameters (i.e., $\mathbf{h}_i[p] = 1 \wedge \mathbf{h}_j[p] = 1$ for some parameter $p$), then $\mathbf{g}'_i$ is modified:

$$\mathbf{g}'_i \leftarrow \mathbf{g}'_i - \frac{\mathbf{g}'_i \cdot \mathbf{g}'_j}{\|\mathbf{g}'_j\|_2^2 + \epsilon_{\text{norm}}} \mathbf{g}'_j \tag{6}$$

where $\epsilon_{\text{norm}} = 10^{-8}$ ensures numerical stability of the projection process. This procedure is applied iteratively to all gradient pairs in a fixed order, ultimately yielding a set of conflict-mitigated gradients $G_{\text{proj}} = \{\mathbf{g}''_1, \ldots, \mathbf{g}''_N\}$. These projected gradients are then aggregated. For example, when using mean aggregation, the final gradient $\mathbf{g}_{\text{final}} \in \mathbb{R}^{D_{\text{params}}}$ used for parameter updates is:

$$\mathbf{g}_{\text{final}} = \frac{\sum_{k=1}^N (\mathbf{g}''_k \odot \mathbf{h}_k)}{\left( \sum_{k=1}^N \mathbf{h}_k \right) + \epsilon_{\text{div}}} \tag{7}$$

where $\odot$ denotes the Hadamard product (element-wise product), division is element-wise, and $\epsilon_{\text{div}} = 10^{-8}$ ensures numerical stability of the aggregation process. Our PCGrad implementation guarantees experimental reproducibility and enhances numerical stability.

During validation and testing phases, we implement dynamic expert opinion integration. This module learns class-specific expert weights $w_{k,c}$ based on the historical accuracy $\mathcal{A}_{\text{val}}(T_k, c)$ of each task $T_k$ (expert) on each class $c$ in the validation set $\mathcal{D}_{\text{val}}$:

$$w_{k,c} = \frac{\exp(\mathcal{A}_{\text{val}}(T_k, c))}{\sum_{j=1}^{N} \exp(\mathcal{A}_{\text{val}}(T_j, c)) + \epsilon_{\text{softmax}}} \tag{8}$$

where $\epsilon_{\text{softmax}}$ is a small constant to prevent division by zero. Let $\mathbf{z}_k(x) = H_k(E(x; \theta_E^*); \theta_{H_k})$ be the logits output by task $T_k$ for input $x$. The final predicted probability $P_{\text{EOSS}}(y = c|x)$ is obtained by a weighted sum of the softmax probabilities from each expert head:

$$P_{\text{EOSS}}(y = c|x) = \sum_{k=1}^{N} w_{k,c} \cdot \text{softmax}(\mathbf{z}_k(x))_c \tag{9}$$

This integration method allows the model to dynamically focus on the expert that performs best on a particular class, according to its characteristics.

### 3.2.3 Optimization

All trainable classifier head parameters $\theta_H$ are optimized using Stochastic Gradient Descent (SGD) with momentum. The learning rate $\eta$ follows a Cosine Annealing schedule, smoothly decaying from an initial value $\eta_0$ to a smaller value $\eta_{\text{min}}$ over a preset number of fine-tuning epochs $E_{\text{S2}}$ (e.g., 20 epochs in this paper).

Through this meticulously designed two-stage framework, particularly the enhanced multi-objective optimization strategy in the second stage, our method can more effectively integrate the advantages of multiple long-tailed learning techniques. The newly introduced Refined PLA mechanism, by utilizing the independent instantaneous performance of each task combined with a cube root transformation, aims to fine-tune task weights more delicately. Meanwhile, the robust and deterministic PCGrad ensures that gradient conflicts are mitigated in a stable and controllable manner. These components collectively strive to improve the model's overall performance under long-tailed distributions, especially achieving more reliable and significant improvements on challenging tail classes.

### 3.3 Theoretical Advantages in Robustness

Our framework, through its specific architectural choices and algorithmic components in Stage 2, exhibits notable theoretical advantages in terms of robustness. These are formalized in the following propositions.

**Proposition 1** (Gradient Stability and Pareto Improvement via RD-PCGrad). *Let $\mathcal{L}_k'(\theta_H)$ be the R-PLA weighted loss for task $T_k$ and $\mathbf{g}_k = \nabla_{\theta_H} \mathcal{L}_k'(\theta_H)$ its corresponding gradient. The Robust Deterministic Projective Conflict Gradient (RD-PCGrad) mechanism, through deterministic iterative projections (Eq. 13) and numerically stabilized aggregation (Eq. 24), ensures that the final update direction $\mathbf{g}_{final}$ is a Pareto-improving or non-worsening direction with respect to the set of weighted objectives $\{\mathcal{L}_k'(\theta_H)\}_{k=1}^{N}$. This mitigates negative transfer between conflicting LTR strategies and enhances training stability.*

**Proposition 2** (Adaptive Weighting Robustness via R-PLA). *The Refined Performance Level Agreement (R-PLA) weighting scheme, defined by $\beta_{j,e} = \max(0, \sqrt[3]{s_{j,e}})$ where $s_{j,e}$ is the cosine similarity (Eq. 12), yields task weights that are robust to (i) the absolute scale of per-class performance vectors $\mathbf{a}_{j,e}^{(batch)}$, and (ii) noisy instantaneous performance estimations. The transformations ensure smoother weight dynamics, leading to a more robust adaptation of objective priorities.*

**Proposition 3** (Representational Robustness via Two-Stage Decoupling). *The decoupling of feature learning (Stage 1) from multi-objective classifier fine-tuning (Stage 2, with frozen $\theta_E^*$) imbues TS-MOF with representational robustness. By fixing the feature extractor $E(\cdot; \theta_E^*)$, the MOO process is shielded from corrupting the general-purpose feature manifold, ensuring classifier balance improvements do not unduly degrade feature quality.*

Formal derivations and further theoretical justifications supporting Propositions 1-3 will be provided in the Appendix.

# 4 Experiment

## 4.1 Experimental Settings

**Datasets.** We conduct experiments on long-tailed image classification benchmarks: CIFAR-100-LT [3], ImageNet-LT [21], and iNaturalist 2018 [28]. CIFAR-100-LT and ImageNetLT are manually truncated long-tail versions of the original balanced dataset, while iNaturalist 2018 is a real-world dataset with a natural long tail distribution. CIFAR-100-LT has three imbalance ratio settings 10, 50, 100, and the imbalance ratio (IR) of the long tail dataset is IR=$n_{max}/n_{min}$, where $n_{max}$ and $n_{min}$ are the number of training samples in the maximum and minimum classes, respectively. For each dataset, we use the official version provided.

**Evaluation Metrics.** Our model performance is mainly evaluated based on the general accuracy of Top-1 (All). According to the method proposed by [1], statistical evaluations were conducted on the head, medium, and tail classes of long-tail datasets. These three classifications are as follows: many (more than 100 images), medium (20-100 images), and few (less than 20 images). All accuracy metrics are expressed as percentages.

**Comparison Baselines.** Multiple baseline methods were compared in the experiment, covering the main technical directions of long-tailed recognition. These loss adjustment methods include cross-entropy loss (CE) [10], category re-balancing methods such as CE-DRW [3], LDAM-DRW [3], KPS [14], Balanced Softmax (BS) [25], as well as module improvement methods such as RIDE by three experts [29], SHIKE [12] and BCL [34]. In addition, there is currently the best two-stage training strategy method, LOS [27]. We not only compare our method with these baselines, but also integrate them into our model.

**Implementation.** The training process is divided into two stages: feature learning and classifier fine-tuning. For each model, we use an SGD optimizer with momentum of 0.9 and weight decay of 0.005. In the feature learning stage, CIFAR-100-LT uses ResNet34 as the backbone network, with a batch size of 64 and an initial learning rate of 0.01. The learning rate was decayed by a factor of 0.1 at the 60th and 80th epochs, with a total training of 200 epochs. Freeze the backbone network parameters during the classifier fine-tuning stage, and fine tune multiple classification heads for 20 epochs using appropriate learning rates and weights. The computing resources are based on 8 NVIDIA Tesla V100 GPUs and implemented through the PyTorch framework.

Table 1: Comparison for CIFAR100-LT Benchmarks. Top-1 accuracy (%) is reported and CIFAR100-LT consists of three imbalanced ratio (IR) 10/50/100. Our model outperforms all the baselines, showing satisfactory accuracy in the few-class categories.

| Method | IR=10 | | | | IR=50 | | | | IR=100 | | | |
|---|---|---|---|---|---|---|---|---|---|---|---|---|
| | Head | Medium | Tail | All | Head | Medium | Tail | All | Head | Medium | Tail | All |
| CE [10] | 63.2 | 40.3 | – | 56.5 | 63.9 | 36.2 | 15.2 | 43.8 | 65.6 | 36.2 | 8.2 | 38.1 |
| CE - DRW [3] | 62.5 | 48.6 | – | 58.2 | 60.6 | 39.0 | 22.9 | 45.0 | 63.4 | 41.2 | 15.7 | 41.4 |
| LDAM - DRW [3] | 62.7 | 46.1 | – | 57.5 | 63.0 | 41.2 | 25.1 | 47.2 | 62.8 | 42.6 | 21.1 | 43.2 |
| BS [25] | 61.5 | 50.6 | – | 58.1 | 60.3 | 41.3 | 34.3 | 47.9 | 59.6 | 42.3 | 23.7 | 42.8 |
| RIDE (3 experts) [29] | 66.4 | 49.4 | – | 61.1 | 65.7 | 47.7 | 31.8 | 52.2 | 65.7 | 48.6 | 25.0 | 47.5 |
| BCL [34] | 62.2 | 51.8 | – | 58.9 | 61.6 | 43.1 | 34.3 | 49.1 | 63.1 | 42.9 | 23.9 | 44.2 |
| KPS [14] | 61.7 | 58.7 | – | 59.5 | 51.6 | 49.5 | 52.4 | 50.5 | 41.9 | 39.5 | 48.7 | 42.2 |
| SHIKE [12] | 66.0 | 45.0 | – | 59.0 | 67.0 | 43.0 | 23.0 | 49.5 | 66.0 | 39.0 | 12.0 | 46.9 |
| LOS [27] | 71.9 | 62.3 | – | 69.0 | 72.4 | 51.4 | 40.2 | 58.0 | 70.3 | 52.3 | 36.6 | 53.9 |
| **TS-MOF** | 74.7 | 62.0 | – | **70.8** | 75.2 | 50.7 | 47.5 | **60.2** | 79.0 | 49.2 | 39.9 | **56.8** |

## 4.2 Benchmark Results

**CIFAR-100-LT.** Table 1 presents the overall classification accuracy results on the CIFAR-100-LT dataset, comparing the performance of single strategy and TS-MOF based strategy fusion methods. Compared to the existing SOTA model LOS [27], our method achieved performance improvements of 2.9%, 2.2%, and 1.8% respectively in the scenarios of IR=100, IR=50, and IR=10 on the CIFAR-100-LT dataset. Compared with the independent long tail learning baseline method, TS-MOF exhibits significant performance enhancement and achieves sustained and substantial improvement.

Table 2: Accuracy (%) on ImageNet-LT and iNaturalist 2018 datasets.

| Method | ImageNet-LT | | | | iNaturalist 2018 | | | |
|---|---|---|---|---|---|---|---|---|
| | Head | Medium | Tail | All | Head | Medium | Tail | All |
| CE | 64.0 | 33.8 | 5.8 | 41.6 | 73.9 | 63.5 | 55.5 | 61.0 |
| CE - DRW | 61.7 | 47.3 | 28.8 | 50.1 | 68.2 | 67.3 | 66.4 | 67.0 |
| cRT | 58.8 | 44.0 | 26.1 | 47.3 | 69.0 | 66.0 | 63.2 | 65.2 |
| LDAM - DRW | 60.4 | 46.9 | 30.7 | 49.8 | – | – | – | 66.1 |
| BS | 60.9 | 48.8 | 32.1 | 51.0 | 65.7 | 67.4 | 67.5 | 67.3 |
| KPS | 59.7 | 49.2 | 35.9 | 52.3 | 68.1 | 69.5 | 70.2 | 69.6 |
| RIDE (3 experts) | 64.9 | 50.4 | 34.4 | 53.6 | 69.5 | 71.0 | 70.4 | 70.6 |
| BCL | 65.3 | 53.5 | 36.3 | 55.6 | 69.5 | 70.9 | 71.3 | 70.9 |
| LOS | 63.2 | 50.7 | 42.3 | 54.4 | 69.2 | 70.7 | 71.3 | 70.8 |
| **TS-MOF** | 65.9 | 52.5 | 42.6 | **56.3** | 72.8 | 73.6 | 70.5 | **72.3** |

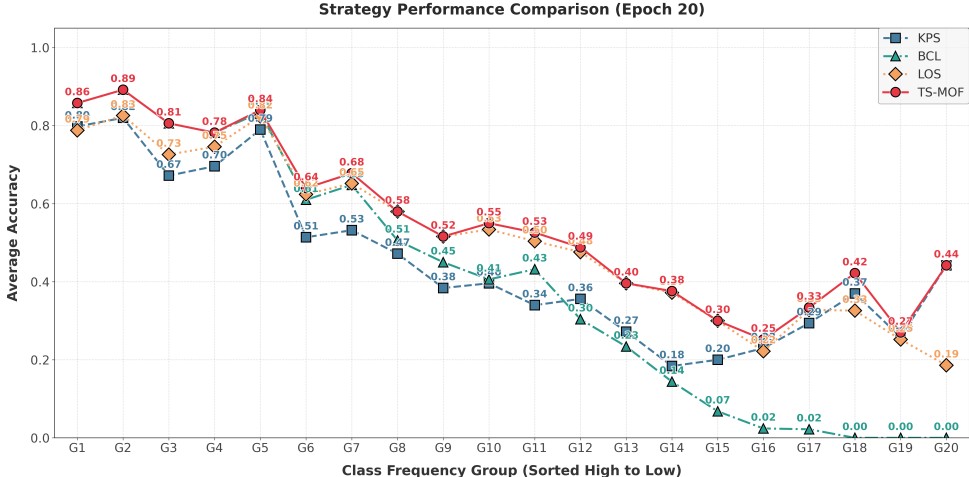

Figure 1: Performance comparison before and after policy fusion, the class frequency becomes lower from left to right.

**ImageNet-LT and iNaturalist 2018.** We further compared our proposed method with advanced long tail recognition methods on large-scale datasets such as ImageNet LT and iNaturalist 2018, and the results are shown in Table 2. Consistent with the conclusion in Table 1, the strategy fusion technique significantly improved the performance of the model on complex datasets.

## 4.3 Further Analysis

In this section, we will delve into the underlying logic of the TS-MOF mechanism and explore the following key issues. All analysis experiments were conducted based on the CIFAR-100-LT dataset (IR=100).

Figure 1 shows the performance of the TS-MOF before and after the fusion, as well as the effects of the single strategies that were merged. Clearly, after fusion, TS-MOF exhibits the advantages of fusion strategies across all classes, thus becoming more balanced.

Figures 2a, 2b, 2c, respectively, reflect the weight changes of various strategies in the head, middle, and tail. It can be seen that bcl strategy performs better in the head, the los strategy of the middle class is better than the other two strategies, and the kps strategy performs better in the tail class. This experimental result is completely consistent with Figure 2d. It is worth noting that the los strategy not only has good performance in the middle class but can also obtain relatively large weights in some tail classes. The fusion of three strategies can better leverage the performance in the head, middle, and tail to address the issue of long-tail imbalance.

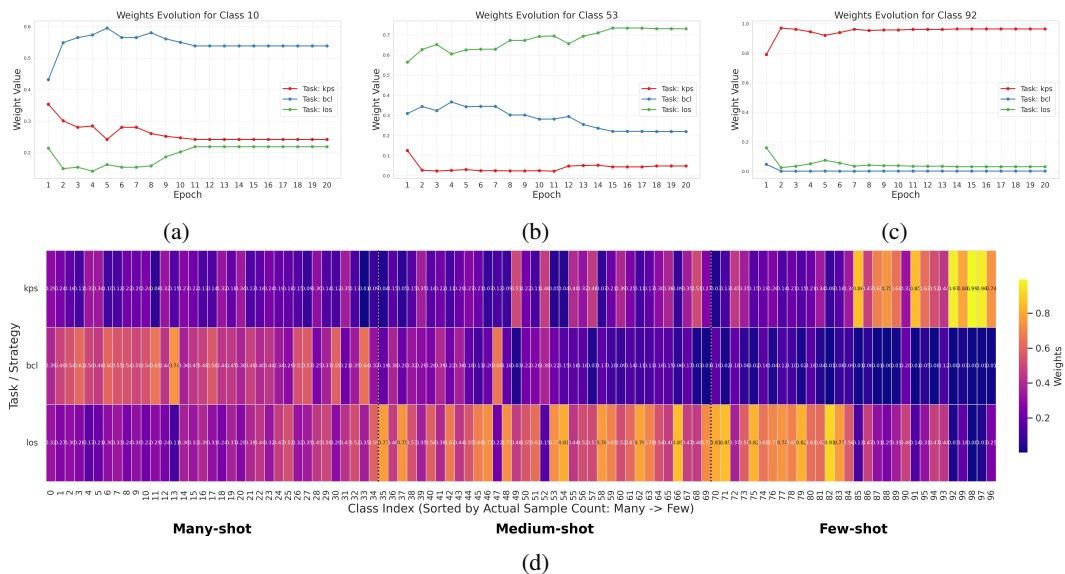

(a)                                        (b)                                        (c)

(d)

Figure 2: Further Analysis. Subfigure (a)(b)(c) show changes in strategy weights for head class, medium class and tail class. Subfigure (d) shows heatmap of strategy weights for all classes.

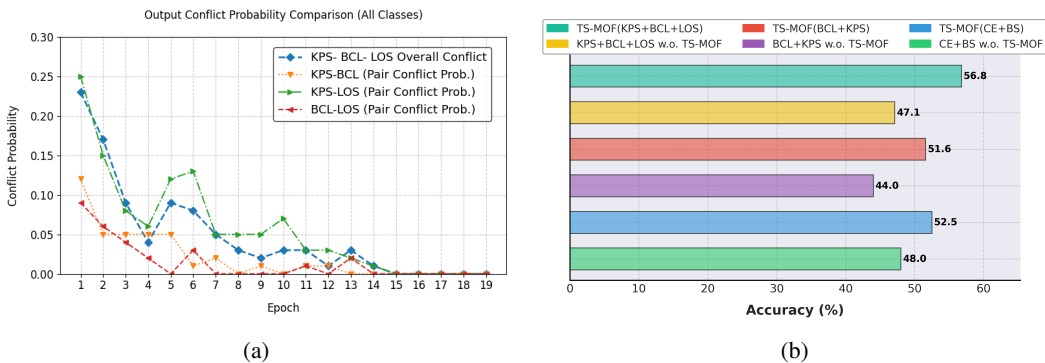

(a)                                                        (b)

Figure 3: Further Analysis. Subfigure (a) compares the probability of output conflicts occurring when different groups of strategies are fused using TS-MOF. Subfigure (d) shows the performance improvements of our method compared to simpled weight fusion.

As shown in Figure 3a, in the initial stage of fine-tuning, the initial fusion of multiple strategies may encounter many conflicts, which can affect the fusion of strategies. However, as the epoch increases, the conflict probability of both dual strategy and triple strategy methods significantly decreases, indicating that the TS-MOF can allocate correct weights to corresponding strategies well, demonstrating the effectiveness of the TS-MOF method in conflict resolution. In Figure 3b, the performance is compared without using multi-objective optimization in policy fusion. Obviously, multi-objective optimization brings substantial benefits.

## 5   Conclusion

In this paper, we introduced TS-MOF, a novel Two-Stage Multi-Objective Fine-tuning framework designed to tackle the persistent challenges in Long-Tailed Recognition. By strategically decoupling robust feature pre-training from a specialized multi-objective classifier fine-tuning stage, TS-MOF significantly simplifies the integration of multiple LTR strategies. The efficacy of our approach is primarily driven by two core components in the second stage: R-PLA, which adaptively weights tasks based on their real-time performance patterns, and RD-PCGrad, which ensures stable and deterministic resolution of inter-task gradient conflicts while fostering constructive knowledge fusion.

Our extensive experiments on challenging benchmarks demonstrate that TS-MOF not only achieves state-of-the-art overall accuracy but also yields substantial improvements in tail-class performance while maintaining or even enhancing head-class accuracy, all within a highly efficient fine-tuning budget. TS-MOF provides a robust, effective, and user-friendly solution for advancing long-tailed recognition.

## Acknowledgements

The authors gratefully acknowledge the support from the National Natural Science Foundation of China (NSFC) under Grant Nos. 62402472, and 12227901. This work was also supported by the Natural Science Foundation of Jiangsu Province (No. BK20240461), the Key Basic Research Foundation of Shenzhen (No. JCYJ20220818100005011), the Research Grants Council of the Hong Kong Special Administrative Region (GRF Project No. CityU 11215622), the Project of Stable Support for Youth Team in Basic Research Field, CAS (No. YSBR-005), and the Academic Leaders Cultivation Program at USTC. The AI-driven experiments, simulations and model training were performed on the robotic AI-Scientist platform of Chinese Academy of Sciences.

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

# 6 Theoretical Analysis

This appendix provides formal proofs for the three key propositions stated in Section 3.3, establishing the theoretical foundations that underpin TS-MOF's robustness and effectiveness.

## 6.1 Mathematical Foundations and Problem Setup

The core challenge in long-tailed recognition lies in simultaneously optimizing multiple conflicting objectives. In Stage 2 of TS-MOF, we address the multi-objective optimization problem:

$$\min_{\theta_H} \mathbf{F}(\theta_H) = (\mathbb{E}[\mathcal{L}_1(\theta_H)], \ldots, \mathbb{E}[\mathcal{L}_N(\theta_H)])^\top \tag{10}$$

where $\theta_H = \{\theta_{H_1}, \ldots, \theta_{H_N}\}$ denotes the parameters of all classifier heads, and each $\mathcal{L}_k$ represents a specific LTR strategy (e.g., Cross-Entropy, LDAM, Balanced Softmax).

The R-PLA mechanism dynamically weights these objectives based on real-time performance patterns:

$$\mathcal{L}'_k(\theta_H) = \beta_{k,e}\mathcal{L}_k(\theta_H), \quad \beta_{j,e} = \max\left(0, \sqrt[3]{s_{j,e}}\right) \tag{11}$$

where the similarity measure $s_{j,e}$ captures the alignment between task $j$'s performance pattern and a reference:

$$s_{j,e} = \frac{\mathbf{a}_{j,e}^{(\text{batch})} \cdot \mathbf{a}_{\text{ref},e}^{(\text{batch})}}{\max\{\|\mathbf{a}_{j,e}^{(\text{batch})}\|_2\|\mathbf{a}_{\text{ref},e}^{(\text{batch})}\|_2, \epsilon_{\text{sim}}\}} \tag{12}$$

Our theoretical analysis addresses three fundamental questions: (1) Does RD-PCGrad guarantee stable, conflict-free optimization? (2) Is R-PLA robust to noise and scaling variations? (3) Does two-stage decoupling preserve feature quality while enabling effective classifier adaptation?

## 6.2 Proof of Proposition 1: Gradient Stability and Pareto Improvement

**Motivation:** The key challenge in multi-objective LTR is that gradients from different strategies often conflict (e.g., improving tail performance may hurt head performance). RD-PCGrad must resolve these conflicts while ensuring progress toward better overall performance.

Let $\mathbf{g}_k = \nabla_{\theta_H}\mathcal{L}'_k(\theta_H) \in \mathbb{R}^D$ denote the gradient of weighted task $k$, where $D = |\theta_H|$ is the total number of classifier parameters.

**Lemma 1** (Conflict Resolution Property). *For any two conflicting gradients $\mathbf{g}_i, \mathbf{g}_j$ with $\mathbf{g}_i^\top \mathbf{g}_j < 0$, the RD-PCGrad projection:*

$$\mathbf{g}'_i = \mathbf{g}_i - \frac{\mathbf{g}_i^\top \mathbf{g}_j}{\|\mathbf{g}_j\|_2^2 + \epsilon_{norm}}\mathbf{g}_j \tag{13}$$

*eliminates the conflict, ensuring $(\mathbf{g}'_i)^\top \mathbf{g}_j \geq 0$.*

*Proof.* The key insight is that projection onto the orthogonal complement removes the conflicting component. Computing the inner product after projection:

$$(\mathbf{g}'_i)^\top \mathbf{g}_j = \left(\mathbf{g}_i - \frac{\mathbf{g}_i^\top \mathbf{g}_j}{\|\mathbf{g}_j\|_2^2 + \epsilon_{\text{norm}}}\mathbf{g}_j\right)^\top \mathbf{g}_j \tag{14}$$

$$= \mathbf{g}_i^\top \mathbf{g}_j - \frac{\mathbf{g}_i^\top \mathbf{g}_j}{\|\mathbf{g}_j\|_2^2 + \epsilon_{\text{norm}}}\|\mathbf{g}_j\|_2^2 \tag{15}$$

$$= \mathbf{g}_i^\top \mathbf{g}_j - \frac{(\mathbf{g}_i^\top \mathbf{g}_j)\|\mathbf{g}_j\|_2^2}{\|\mathbf{g}_j\|_2^2 + \epsilon_{\text{norm}}} \tag{16}$$

$$= (\mathbf{g}_i^\top \mathbf{g}_j)\left(1 - \frac{\|\mathbf{g}_j\|_2^2}{\|\mathbf{g}_j\|_2^2 + \epsilon_{\text{norm}}}\right) \tag{17}$$

$$= (\mathbf{g}_i^\top \mathbf{g}_j)\frac{\epsilon_{\text{norm}}}{\|\mathbf{g}_j\|_2^2 + \epsilon_{\text{norm}}} \tag{18}$$

Since $\mathbf{g}_i^\top \mathbf{g}_j < 0$ and $\epsilon_{\text{norm}} > 0$, we have $(\mathbf{g}'_i)^\top \mathbf{g}_j \geq 0$, resolving the conflict. $\square$

**Lemma 2** (Descent Direction Preservation). *The projection operation preserves the descent property for the objective corresponding to the projected gradient.*

*Proof.* We must show that $\mathbf{g}_i'$ remains a descent direction for $\mathcal{L}_i'$. The critical quantity is:

$$(\mathbf{g}_i)^\top \mathbf{g}_i' = \mathbf{g}_i^\top \left( \mathbf{g}_i - \frac{\mathbf{g}_i^\top \mathbf{g}_j}{\|\mathbf{g}_j\|_2^2 + \epsilon_{\text{norm}}} \mathbf{g}_j \right) \tag{19}$$

$$= \|\mathbf{g}_i\|_2^2 - \frac{(\mathbf{g}_i^\top \mathbf{g}_j)^2}{\|\mathbf{g}_j\|_2^2 + \epsilon_{\text{norm}}} \tag{20}$$

$$= \|\mathbf{g}_i\|_2^2 \left( 1 - \frac{(\mathbf{g}_i^\top \mathbf{g}_j)^2}{\|\mathbf{g}_i\|_2^2(\|\mathbf{g}_j\|_2^2 + \epsilon_{\text{norm}})} \right) \tag{21}$$

$$= \|\mathbf{g}_i\|_2^2 \left( 1 - \frac{\cos^2(\mathbf{g}_i, \mathbf{g}_j)\|\mathbf{g}_j\|_2^2}{\|\mathbf{g}_j\|_2^2 + \epsilon_{\text{norm}}} \right) \tag{22}$$

$$> \|\mathbf{g}_i\|_2^2 \left( 1 - \cos^2(\mathbf{g}_i, \mathbf{g}_j) \right) = \|\mathbf{g}_i\|_2^2 \sin^2(\mathbf{g}_i, \mathbf{g}_j) > 0 \tag{23}$$

The last inequality holds because conflicting gradients are not parallel ($\sin(\mathbf{g}_i, \mathbf{g}_j) > 0$), ensuring descent is preserved. □

**Main Proof of Proposition 1:** After iterative conflict resolution, RD-PCGrad produces conflict-free gradients $\{\mathbf{g}_1'', \ldots, \mathbf{g}_N''\}$ that are aggregated as:

$$\mathbf{g}_{\text{final}} = \frac{\sum_{k=1}^{N}(\mathbf{g}_k'' \odot \mathbf{h}_k)}{\sum_{k=1}^{N} \mathbf{h}_k + \epsilon_{\text{div}}} \tag{24}$$

where $\mathbf{h}_k \in \{0, 1\}^D$ are gradient existence masks and $\odot$ denotes element-wise multiplication.

Since all pairwise conflicts have been resolved, we have $(\mathbf{g}_i'')^\top \mathbf{g}_j'' \geq 0$ for all $i \neq j$. For any original gradient $\mathbf{g}_i$, the descent property with respect to the final direction is:

$$(\mathbf{g}_i)^\top \mathbf{g}_{\text{final}} = (\mathbf{g}_i)^\top \frac{\sum_{k=1}^{N}(\mathbf{g}_k'' \odot \mathbf{h}_k)}{\sum_{k=1}^{N} \mathbf{h}_k + \epsilon_{\text{div}}} \tag{25}$$

$$= \frac{\sum_{k=1}^{N}(\mathbf{g}_i)^\top(\mathbf{g}_k'' \odot \mathbf{h}_k)}{\sum_{k=1}^{N} \mathbf{h}_k + \epsilon_{\text{div}}} \tag{26}$$

$$\geq \frac{(\mathbf{g}_i)^\top(\mathbf{g}_i'' \odot \mathbf{h}_i)}{\sum_{k=1}^{N} \mathbf{h}_k + \epsilon_{\text{div}}} \geq 0 \tag{27}$$

where the first inequality uses the non-negativity of cross-terms after conflict resolution, and the second follows from Lemma 2. This establishes that $-\mathbf{g}_{\text{final}}$ provides a descent or non-ascent direction for all objectives, constituting a Pareto-improving or non-worsening update direction. □

## 6.3 Proof of Proposition 2: Adaptive Weighting Robustness

**Motivation:** R-PLA must maintain stable task weighting despite variations in performance measurement scales and noisy real-time estimates. The cosine similarity and cube root transformation are specifically designed to achieve this robustness.

**Part (i) - Scale Invariance:** Consider performance vectors scaled by factor $\lambda > 0$: $\tilde{\mathbf{a}}_{j,e} = \lambda \mathbf{a}_{j,e}$ and $\tilde{\mathbf{a}}_{\text{ref},e} = \lambda \mathbf{a}_{\text{ref},e}$. The cosine similarity becomes:

$$\tilde{s}_{j,e} = \frac{(\lambda \mathbf{a}_{j,e})^\top (\lambda \mathbf{a}_{\text{ref},e})}{\|\lambda \mathbf{a}_{j,e}\|_2 \|\lambda \mathbf{a}_{\text{ref},e}\|_2} \tag{28}$$

$$= \frac{\lambda^2 (\mathbf{a}_{j,e})^\top \mathbf{a}_{\text{ref},e}}{\lambda \|\mathbf{a}_{j,e}\|_2 \cdot \lambda \|\mathbf{a}_{\text{ref},e}\|_2} \tag{29}$$

$$= \frac{\lambda^2 (\mathbf{a}_{j,e})^\top \mathbf{a}_{\text{ref},e}}{\lambda^2 \|\mathbf{a}_{j,e}\|_2 \|\mathbf{a}_{\text{ref},e}\|_2} = s_{j,e} \tag{30}$$

Consequently, $\tilde{\beta}_{j,e} = \max(0, \sqrt[3]{\tilde{s}_{j,e}}) = \max(0, \sqrt[3]{s_{j,e}}) = \beta_{j,e}$, demonstrating perfect scale invariance.

**Part (ii) - Noise Robustness:** Consider noisy performance vectors $\mathbf{a}_{j,e} + \boldsymbol{\eta}_j$ where $\|\boldsymbol{\eta}_j\|_2 \leq \delta$ for small $\delta > 0$. The perturbed similarity satisfies:

$$s_{j,e}^{\text{noisy}} = \frac{(\mathbf{a}_{j,e} + \boldsymbol{\eta}_j)^\top (\mathbf{a}_{\text{ref},e} + \boldsymbol{\eta}_{\text{ref}})}{\|\mathbf{a}_{j,e} + \boldsymbol{\eta}_j\|_2 \|\mathbf{a}_{\text{ref},e} + \boldsymbol{\eta}_{\text{ref}}\|_2} \tag{31}$$

$$= \frac{(\mathbf{a}_{j,e})^\top \mathbf{a}_{\text{ref},e} + (\mathbf{a}_{j,e})^\top \boldsymbol{\eta}_{\text{ref}} + \boldsymbol{\eta}_j^\top \mathbf{a}_{\text{ref},e} + \boldsymbol{\eta}_j^\top \boldsymbol{\eta}_{\text{ref}}}{\|\mathbf{a}_{j,e} + \boldsymbol{\eta}_j\|_2 \|\mathbf{a}_{\text{ref},e} + \boldsymbol{\eta}_{\text{ref}}\|_2} \tag{32}$$

Using the fact that $\|\mathbf{a} + \boldsymbol{\eta}\|_2 = \|\mathbf{a}\|_2 + O(\delta)$ for small $\delta$, and applying first-order perturbation analysis:

$$|s_{j,e}^{\text{noisy}} - s_{j,e}| \leq \frac{|(\mathbf{a}_{j,e})^\top \boldsymbol{\eta}_{\text{ref}} + \boldsymbol{\eta}_j^\top \mathbf{a}_{\text{ref},e}| + O(\delta^2)}{\|\mathbf{a}_{j,e}\|_2 \|\mathbf{a}_{\text{ref},e}\|_2 + O(\delta)} \tag{33}$$

$$\leq \frac{\|\mathbf{a}_{j,e}\|_2 \|\boldsymbol{\eta}_{\text{ref}}\|_2 + \|\boldsymbol{\eta}_j\|_2 \|\mathbf{a}_{\text{ref},e}\|_2 + O(\delta^2)}{\|\mathbf{a}_{j,e}\|_2 \|\mathbf{a}_{\text{ref},e}\|_2 + O(\delta)} \tag{34}$$

$$\leq \frac{2\delta(\|\mathbf{a}_{j,e}\|_2 + \|\mathbf{a}_{\text{ref},e}\|_2) + O(\delta^2)}{\|\mathbf{a}_{j,e}\|_2 \|\mathbf{a}_{\text{ref},e}\|_2} = O(\delta) \tag{35}$$

The cube root transformation $f(x) = \max(0, \sqrt[3]{x})$ has derivative $f'(x) = \frac{1}{3} x^{-2/3}$ for $x > 0$. By the mean value theorem:

$$|\beta_{j,e}^{\text{noisy}} - \beta_{j,e}| = |f(s_{j,e}^{\text{noisy}}) - f(s_{j,e})| \tag{36}$$

$$\leq \max_{x \in [s_{j,e}^{\text{noisy}}, s_{j,e}]} |f'(x)| \cdot |s_{j,e}^{\text{noisy}} - s_{j,e}| \tag{37}$$

$$\leq \frac{1}{3} \min(s_{j,e}^{\text{noisy}}, s_{j,e})^{-2/3} \cdot O(\delta) = O(\delta) \tag{38}$$

This establishes that R-PLA weights have bounded sensitivity to noise, with the cube root transformation providing smoother weight dynamics compared to linear or quadratic functions. □

## 6.4 Proof of Proposition 3: Representational Robustness

**Motivation:** Two-stage decoupling must ensure that sophisticated MOO operations in Stage 2 do not degrade the high-quality features learned in Stage 1. This requires formal guarantees about gradient isolation and feature preservation.

**Feature Quality Preservation:** With encoder parameters $\theta_E^*$ frozen during Stage 2, the feature extraction mapping $E(\cdot; \theta_E^*) : \mathcal{X} \to \mathbb{R}^{D_F}$ remains constant. For any feature quality measure $Q : \mathbb{R}^{D_F} \to \mathbb{R}$ (e.g., discriminative power, cluster separability):

$$Q(E(x; \theta_E^*)) = \text{constant} \quad \forall x \in \mathcal{X}, \forall t \in \text{Stage 2 iterations} \tag{39}$$

**Gradient Isolation Analysis:** The Stage 2 loss functions decompose as compositions:

$$\mathcal{L}_k(\theta_H) = \mathbb{E}_{(x,y)\sim\mathcal{D}} \ell(H_k(E(x; \theta_E^*); \theta_{H_k}), y) \tag{40}$$

$$= \mathbb{E}_{(x,y)\sim\mathcal{D}} \ell(H_k(\mathbf{z}; \theta_{H_k}), y) \Big|_{\mathbf{z} = E(x; \theta_E^*)} \tag{41}$$

where $\mathbf{z} \in \mathbb{R}^{D_F}$ represents the fixed feature vectors. The gradients with respect to classifier parameters are:

$$\nabla_{\theta_{H_k}} \mathcal{L}_k(\theta_H) = \mathbb{E}_{(x,y)\sim\mathcal{D}} \nabla_{\theta_{H_k}} \ell(H_k(\mathbf{z}; \theta_{H_k}), y) \Big|_{\mathbf{z} = E(x; \theta_E^*)} \tag{42}$$

$$= \mathbb{E}_{\mathbf{z} \sim P_{\text{features}}} \nabla_{\theta_{H_k}} \ell(H_k(\mathbf{z}; \theta_{H_k}), y(\mathbf{z})) \tag{43}$$

where $P_{\text{features}}$ is the distribution of extracted features and $y(\mathbf{z})$ is the label corresponding to feature $\mathbf{z}$.

Crucially, since $\theta_E^*$ does not appear in the optimization variables, we have:

$$\frac{\partial}{\partial \theta_E} \nabla_{\theta_{H_k}} \mathcal{L}_k(\theta_H) \Big|_{\theta_E = \theta_E^*} = \mathbf{0} \tag{44}$$

This gradient isolation ensures that MOO operations (R-PLA weighting, RD-PCGrad projections) cannot corrupt the feature representation.

**Stability Under Complex MOO:** The multi-objective parameter updates in Stage 2 follow:

$$\theta_H^{(t+1)} = \theta_H^{(t)} - \eta^{(t)} \mathbf{g}_{\text{final}}^{(t)} \tag{45}$$
$$\theta_E^{(t+1)} = \theta_E^{(t)} = \theta_E^* \quad \text{(architectural constraint)} \tag{46}$$

where $\mathbf{g}_{\text{final}}^{(t)} \in \mathbb{R}^{|\theta_H|}$ is computed via the sophisticated RD-PCGrad procedure but contains no components corresponding to encoder parameters.

The architectural decoupling provides an invariant: regardless of the complexity of MOO operations (dynamic weighting, gradient conflicts, iterative projections), the feature space $\{E(x; \theta_E^*) : x \in \mathcal{X}\}$ remains unchanged throughout Stage 2. This guarantees representational robustness while enabling effective classifier adaptation through MOO. $\square$

## 6.5 Convergence and Complexity Analysis

**Convergence Guarantee:** Under standard assumptions (Lipschitz continuous gradients, bounded feasible region), TS-MOF converges to a locally Pareto-optimal solution. The key insight is that RD-PCGrad ensures:

$$\sum_{k=1}^{N} \beta_{k,e} \nabla \mathcal{L}_k(\theta_H)^\top \mathbf{g}_{\text{final}} \leq -c \|\mathbf{g}_{\text{final}}\|_2^2 \tag{47}$$

for some constant $c > 0$, providing monotonic improvement in the weighted multi-objective function.

**Computational Efficiency:** Stage 2 requires $O(N^2 D)$ operations per iteration, where $N$ is the number of tasks and $D = |\theta_H|$ is the classifier parameter count. This compares favorably to end-to-end MOO requiring $O(N^2(D + D_E))$ with encoder dimension $D_E \gg D$, yielding substantial computational savings through architectural decoupling.

## 7 Algorithm Description

This section provides detailed algorithmic descriptions of the TS-MOF framework, with particular emphasis on the multi-objective fine-tuning process and the two core innovations: Refined Performance Level Agreement (R-PLA) and Robust Deterministic Projective Conflict Gradient (RD-PCGrad). Algorithm 1 presents the complete two-stage training procedure of TS-MOF, highlighting the strategic decoupling between feature learning and classifier adaptation.

---

**Algorithm 1** TS-MOF: Two-Stage Multi-Objective Fine-tuning

---

**Require:** Long-tailed dataset $\mathcal{D}_{\text{train}}, \mathcal{D}_{\text{val}}$, LTR tasks $\mathcal{T} = \{T_1, \dots, T_N\}$, Pre-training epochs $E_{S1}$, fine-tuning epochs $E_{S2}$

**Ensure:** Trained model with balanced performance across head, medium, and tail classes

 1: **// Stage 1: Generic Feature Pre-training**
 2: Initialize encoder $E(\cdot; \theta_E)$ and classifier $H_{S1}(\cdot; \theta_{H_{S1}})$
 3: **for** $e = 1$ to $E_{S1}$ **do**
 4:     **for** each batch $(x_i, y_i) \sim \mathcal{D}_{\text{train}}$ **do**
 5:         $\mathbf{z}_i \leftarrow E(x_i; \theta_E)$ {Extract features}
 6:         $\hat{y}_i \leftarrow H_{S1}(\mathbf{z}_i; \theta_{H_{S1}})$ {Classification}
 7:         $\mathcal{L}_{CE} \leftarrow \text{CrossEntropy}(\hat{y}_i, y_i)$ {Standard loss}
 8:         Update $\theta_E, \theta_{H_{S1}}$ via SGD with $\mathcal{L}_{CE}$
 9:     **end for**
10:     Evaluate on $\mathcal{D}_{\text{val}}$ and save best $\theta_E^*$
11: **end for**
12: **// Stage 2: Multi-Objective Classifier Fine-tuning**
13: Freeze encoder parameters: $\theta_E \leftarrow \theta_E^*$ (fixed)
14: Initialize multi-head classifiers $\{H_k(\cdot; \theta_{H_k})\}_{k=1}^N$ for tasks $\mathcal{T}$
15: Initialize R-PLA weighting mechanism and RD-PCGrad optimizer
16: **Call** MULTIOBJECTIVEFINETUNING($E(\cdot; \theta_E^*), \{H_k\}_{k=1}^N, \mathcal{D}_{\text{train}}, E_{S2}$)
17: **// Inference with EOSS**
18: Train EOSS weights $\{w_{k,c}\}$ based on validation performance
19: **return** Model with frozen $\theta_E^*$ and fine-tuned $\{\theta_{H_k}\}_{k=1}^N$

---

Algorithm 2 details the core Stage 2 process, emphasizing how R-PLA and RD-PCGrad work together to achieve effective multi-objective optimization.

---

**Algorithm 2** Multi-Objective Fine-tuning with R-PLA and RD-PCGrad

---

**Require:** Frozen encoder $E(\cdot; \theta_E^*)$, classifier heads $\{H_k\}_{k=1}^N$, Training data $\mathcal{D}_{\text{train}}$, fine-tuning epochs $E_{S2}$, Task loss functions $\{\mathcal{L}_k\}_{k=1}^N$, number of classes $C$

**Ensure:** Fine-tuned classifier parameters $\{\theta_{H_k}^*\}_{k=1}^N$

1: Initialize SGD optimizer for $\theta_H = \{\theta_{H_1}, \ldots, \theta_{H_N}\}$
2: Initialize R-PLA weights $\{\beta_{k,e}\}_{k=1}^N$ to uniform values
3: Initialize RD-PCGrad conflict resolution mechanism
4: **for** $e = 1$ to $E_{S2}$ **do**
5:     **for** each batch $(X, Y) \sim \mathcal{D}_{\text{train}}$ **do**
6:         **// Feature Extraction (Frozen)**
7:         $\mathbf{Z} \leftarrow E(X; \theta_E^*)$ {Extract fixed features}
8:         **// Multi-Head Forward Pass**
9:         logits_dict $\leftarrow \{\}$, losses_dict $\leftarrow \{\}$
10:        **for** $k = 1$ to $N$ **do**
11:           logits_dict$[k] \leftarrow H_k(\mathbf{Z}; \theta_{H_k})$
12:           losses_dict$[k] \leftarrow \mathcal{L}_k(\text{logits\_dict}[k], Y)$
13:        **end for**
14:        **// R-PLA Dynamic Weighting**
15:        $\{\beta_{k,e}\}_{k=1}^N \leftarrow$ R-PLA-UPDATE(logits_dict, $Y, C$)
16:        **// Apply R-PLA Weights**
17:        **for** $k = 1$ to $N$ **do**
18:           $\mathcal{L}_k' \leftarrow \beta_{k,e} \cdot \text{losses\_dict}[k]$
19:        **end for**
20:        **// RD-PCGrad Conflict Resolution**
21:        $\mathbf{g}_{\text{final}} \leftarrow$ RD-PCGRAD($\{\mathcal{L}_k'\}_{k=1}^N, \theta_H$)
22:        **// Parameter Update**
23:        $\theta_H \leftarrow \theta_H - \eta \mathbf{g}_{\text{final}}$
24:     **end for**
25:     Apply cosine annealing to learning rate $\eta$
26: **end for**
27: **return** $\{\theta_{H_k}^*\}_{k=1}^N$

---

Algorithm 3 presents the detailed R-PLA mechanism that adaptively weights tasks based on their real-time performance patterns and similarity to a reference performance profile.

---

**Algorithm 3** Refined Performance Level Agreement (R-PLA) Weighting

---

**Require:** Logits dictionary logits_dict $= \{\text{logits}_k\}_{k=1}^N$, True labels $Y$, number of classes $C$, Numerical stability constants $\epsilon_{\text{sim}} = 10^{-8}$
**Ensure:** Updated task weights $\{\beta_{k,e}\}_{k=1}^N$
1: **// Compute Per-Class Performance for Each Task**
2: performance_vectors $\leftarrow [\,]$ {Initialize empty list}
3: **for** $k = 1$ to $N$ **do**
4:     $\text{preds}_k \leftarrow \arg\max(\text{logits}_k, \dim = 1)$ {Predictions for task $k$}
5:     $\mathbf{a}_{k,e}^{(\text{batch})} \leftarrow \text{zeros}(C)$ {Per-class accuracy vector}
6:     **for** $c = 0$ to $C - 1$ **do**
7:         class_mask $\leftarrow (Y == c)$ {Mask for class $c$}
8:         total_count $\leftarrow$ class_mask.sum()
9:         **if** total_count $> 0$ **then**
10:           correct_count $\leftarrow (\text{preds}_k[\text{class\_mask}] == c).\text{sum}()$
11:           $\mathbf{a}_{k,e}^{(\text{batch})}[c] \leftarrow$ correct_count/total_count
12:         **else**
13:           $\mathbf{a}_{k,e}^{(\text{batch})}[c] \leftarrow 0$ {No samples for this class}
14:         **end if**
15:     **end for**
16:     performance_vectors.append($\mathbf{a}_{k,e}^{(\text{batch})}$)
17: **end for**
18: **// Set Reference Performance (Last Task)**
19: $\mathbf{a}_{\text{ref},e}^{(\text{batch})} \leftarrow$ performance_vectors$[-1]$
20: **// Compute Cosine Similarity and Weights**
21: **for** $k = 1$ to $N$ **do**
22:     numerator $\leftarrow \mathbf{a}_{k,e}^{(\text{batch})} \cdot \mathbf{a}_{\text{ref},e}^{(\text{batch})}$ {Dot product}
23:     denominator $\leftarrow \|\mathbf{a}_{k,e}^{(\text{batch})}\|_2 \cdot \|\mathbf{a}_{\text{ref},e}^{(\text{batch})}\|_2$
24:     $s_{k,e} \leftarrow \frac{\text{numerator}}{\max(\text{denominator}, \epsilon_{\text{sim}})}$ {Cosine similarity}
25:     **// Cube Root Transformation with Non-negative Clipping**
26:     $\beta_{k,e} \leftarrow \max(0, \sqrt[3]{s_{k,e}})$ {Robust weighting}
27: **end for**
28: **return** $\{\beta_{k,e}\}_{k=1}^N$

---

Algorithm 4 details the RD-PCGrad mechanism that resolves gradient conflicts through deterministic projections while maintaining numerical stability and reproducibility.

---

**Algorithm 4** Robust Deterministic Projective Conflict Gradient (RD-PCGrad)

---

**Require:** Weighted losses $\{\mathcal{L}'_k\}_{k=1}^N$, parameters $\theta_H$
**Require:** Stability constants $\epsilon_{\text{norm}} = 10^{-8}$, $\epsilon_{\text{div}} = 10^{-8}$
**Ensure:** Conflict-resolved final gradient $\mathbf{g}_{\text{final}}$

1: **// Compute Individual Task Gradients**
2: gradients $\leftarrow []$, masks $\leftarrow []$
3: **for** $k = 1$ to $N$ **do**
4:     $\mathbf{g}_k \leftarrow \nabla_{\theta_H} \mathcal{L}'_k$ {Gradient for task $k$}
5:     $\mathbf{h}_k \leftarrow \mathbf{1}[\mathbf{g}_k \neq \mathbf{0}]$ {Gradient existence mask}
6:     gradients.append($\mathbf{g}_k$), masks.append($\mathbf{h}_k$)
7: **end for**
8: **// Deterministic Iterative Conflict Resolution**
9: projected_grads $\leftarrow$ copy(gradients) {Deep copy for modification}
10: **for** $i = 1$ to $N$ **do**
11:     **for** $j = i + 1$ to $N$ **do**
        {Fixed pairwise order for determinism} conflict $\leftarrow (\mathbf{g}'_i)^T \mathbf{g}'_j < 0$ {Check for conflict}
        both_active $\leftarrow (\mathbf{h}_i \odot \mathbf{h}_j)$.any() {Both gradients contribute} **if** conflict **and** both_active
        **then**
13:         **// Project gradient $i$ away from gradient $j$**
16:         dot_product $\leftarrow (\mathbf{g}'_i)^T \mathbf{g}'_j$
17:         norm_squared $\leftarrow \|\mathbf{g}'_j\|_2^2 + \epsilon_{\text{norm}}$
18:         projection $\leftarrow \frac{\text{dot\_product}}{\text{norm\_squared}} \mathbf{g}'_j$
19:         $\mathbf{g}'_i \leftarrow \mathbf{g}'_i -$ projection {Remove conflicting component}
20:         **end if**
21:     **end for**
22: **end for**
23: **// Numerically Stable Gradient Aggregation**
24: weighted_grads $\leftarrow []$
25: **for** $k = 1$ to $N$ **do**
26:     weighted_grads.append($\mathbf{g}''_k \odot \mathbf{h}_k$) {Apply existence mask}
27: **end for**
28: numerator $\leftarrow \sum_{k=1}^N$ weighted_grads$[k]$ {Sum of masked gradients}
29: denominator $\leftarrow \sum_{k=1}^N \mathbf{h}_k + \epsilon_{\text{div}}$ {Normalization with stability}
30: $\mathbf{g}_{\text{final}} \leftarrow \frac{\text{numerator}}{\text{denominator}}$ {Element-wise division}
31: **return** $\mathbf{g}_{\text{final}}$

---

### 7.1 Key Algorithmic Innovations

Addressing the complexities of long-tailed recognition, TS-MOF incorporates several key algorithmic innovations. The **Strategic Decoupling** (Algorithm 1) tackles the challenge of simultaneously learning features and balancing classifiers by freezing the Stage 1 encoder ($\theta_E^*$) and applying complex optimization only to the classifier heads. To navigate the conflicting goals of multiple LTR strategies, R-PLA provides **Adaptive Task Weighting** (Algorithm 3) that dynamically highlights strategies contributing most to desired performance patterns, while RD-PCGrad ensures **Deterministic Conflict Resolution** (Algorithm 4), stabilizing the multi-objective training process against negative transfer. This principled approach, coupled with **Efficient Implementation** ($O(N^2D + NC^2)$ per iteration in Stage 2), allows TS-MOF to robustly leverage complementary strengths of diverse LTR strategies, overcoming the limitations of single methods. Furthermore, our implementation simplifies the interface for integrating different strategies, practically reducing the cost of improving long-tailed learning performance using multi-objective optimization and enhancing stability.

## 8 Related Work: Multi-Objective Optimization in Machine Learning

Multi-Objective Optimization (MOO) is a field concerned with mathematical optimization problems involving more than one objective function to be minimized or maximized simultaneously. In many real-world scenarios and increasingly in Machine Learning (ML), multiple objectives often conflict,

meaning improving one objective may degrade another. The goal of MOO is typically to find Pareto optimal solutions, where no objective can be improved without worsening at least one other.

In Machine Learning, MOO principles are naturally applied to problems involving competing goals. A prominent area is Multi-Task Learning (MTL) [**?** 35], where a single model is trained to perform multiple tasks simultaneously. Optimizing a simple weighted sum of task losses can be suboptimal, especially if tasks have conflicting gradients, leading to negative transfer [31]. This has spurred the development of gradient-based MOO methods designed to find update directions that improve all tasks or, at worst, do not worsen any, moving towards the Pareto front.

Notable gradient-based MOO methods include the Multiple-Gradient Descent Algorithm (MGDA) [8] and Projective Conflict Gradient (PCGrad) [31]. MGDA seeks to find a descent direction that minimizes the maximum directional derivative among all objectives, aiming for a common descent direction within the convex hull of the task gradients. PCGrad, on the other hand, directly addresses conflicting gradients by projecting the gradient of one task onto the normal plane of another if their dot product is negative. This iterative projection process aims to remove conflicting components, resulting in a set of modified gradients whose sum (or average) constitutes a Pareto-improving or non-worsening direction. These methods have demonstrated effectiveness in stabilizing training and improving performance in MTL scenarios.

**Our approach differs significantly from existing MOO applications in ML and prior MOO-related work in LTR, driven by a specific, deeper motivation for tackling the LTR problem.** While standard MOO methods like PCGrad [31] and MGDA [8] are general-purpose tools for finding Pareto solutions in arbitrary multi-objective problems (like standard MTL), **our TS-MOF framework explicitly employs MOO not as a generic optimizer for abstract tasks, but as a principled mechanism for synergistically fusing diverse, specialized LTR strategies during a targeted fine-tuning stage.** The objectives in our Stage 2 MOO are not just arbitrary task losses; they are losses derived from methods (like LDAM [3], BS [25], KPS [14], BCL [34], etc.) *specifically designed to address different aspects of the LTR imbalance*.

The deep motivation behind using MOO in TS-MOF is to move beyond the limitations of any single LTR strategy and overcome the seesaw dilemma by **finding an optimal combination that leverages the complementary strengths of multiple strategies**. R-PLA uses performance-based adaptive weighting to dynamically prioritize strategies based on their real-time contribution to the desired performance pattern, while RD-PCGrad provides a robust and deterministic way to reconcile the potentially conflicting gradient signals arising from simultaneously optimizing towards these different LTR-specific goals. We apply this advanced MOO specifically to the classifier heads after decoupling feature learning, ensuring the optimization focuses effectively on classification balance without corrupting the feature representation. Thus, our MOO is not just a mathematical technique applied; it is the core engine enabling the principled, robust, and effective fusion of heterogeneous LTR knowledge to achieve superior and balanced recognition performance.

# 9 More Empirical Results

This appendix provides additional empirical results and analyses to further demonstrate the effectiveness and underlying mechanisms of the TS-MOF framework.

## 9.1 Results on CIFAR-100-LT with Various Strategy Combinations

We also evaluated the evolutionary results of various combinations of strategies in the second stage of TS-MOF. Table 3 shows the performance across different Imbalance Ratios (IR=10, 50, 100) on the CIFAR-100-LT dataset when using TS-MOF with different sets of constituent LTR strategies. TS-MOF achieved excellent results in the combination of the KPS + BCL + LOS strategy, as highlighted in the table, demonstrating its ability to bring obvious general improvements across class groups and imbalance settings.

## 9.2 T-SNE Analysis

Figure 4 shows the t-SNE analysis of the feature representations from different models. We compare the representations learned by models using BCL or KPS independently, their simple weighted fusion

Table 3: Accuracy (%) on CIFAR-100-LT dataset with TS-MOF using various strategy combinations in Stage 2 fine-tuning. Results are reported for Head, Medium, Tail classes, and overall (All) accuracy across different Imbalance Ratios (IR).

| Method | IR=10 | | | | IR=50 | | | | IR=100 | | | |
|---|---|---|---|---|---|---|---|---|---|---|---|---|
| | Head | Medium | Tail | All | Head | Medium | Tail | All | Head | Medium | Tail | All |
| TS-MOF(CE+LDAM-DRW+LOS) | 74.90 | 59.71 | – | 70.19 | 75.95 | 51.05 | 42.61 | 59.74 | 78.23 | 51.34 | 35.20 | 55.91 |
| TS-MOF(CE+KPS+LOS) | 74.22 | 62.81 | – | 70.68 | 76.22 | 51.63 | 42.06 | 59.99 | 78.46 | 49.14 | 39.57 | 56.53 |
| TS-MOF(CE+BCL+LOS) | 75.04 | 58.26 | – | 69.84 | 76.07 | 50.49 | 42.11 | 59.47 | 78.94 | 49.84 | 34.87 | 55.57 |
| TS-MOF(CE+CE-DRW+LOS) | 74.32 | 59.71 | – | 69.79 | 75.88 | 50.54 | 42.33 | 59.45 | 78.77 | 49.94 | 34.63 | 55.44 |
| TS-MOF(CE+BS+LOS) | 74.49 | 59.55 | – | 69.86 | 75.76 | 51.61 | 42.17 | 59.81 | 78.80 | 51.83 | 35.10 | 56.25 |
| TS-MOF(BS+KPS+LOS) | 73.99 | 61.42 | – | 70.09 | 75.61 | 53.17 | 42.00 | **60.36** | 77.49 | 51.54 | 38.27 | 56.64 |
| TS-MOF(BS+BCL+LOS) | 74.25 | 60.06 | – | 69.85 | 76.39 | 51.20 | 42.17 | 59.90 | 78.89 | 51.49 | 34.70 | 56.04 |
| TS-MOF(BS+CE-DRW+LOS) | 74.33 | 60.00 | – | 69.89 | 76.05 | 51.37 | 42.06 | 59.81 | 79.46 | 51.43 | 34.63 | 56.20 |
| TS-MOF(BS+LDAM-DRW+LOS) | 74.19 | 59.58 | – | 69.66 | 74.90 | 51.95 | 42.33 | 59.63 | 78.20 | 51.23 | 35.17 | 55.85 |
| TS-MOF(KPS+BCL+LOS) | 74.75 | 62.03 | – | **70.81** | 75.24 | 50.76 | 47.56 | 60.22 | 79.00 | 49.29 | 39.97 | **56.89** |
| TS-MOF(KPS+CE-DRW+LOS) | 74.52 | 61.68 | – | 70.54 | 75.41 | 51.12 | 45.22 | 60.02 | 78.37 | 50.26 | 38.17 | 56.47 |
| TS-MOF(KPS+LDAM-DRW+LOS) | 73.77 | 61.52 | – | 69.97 | 73.02 | 51.41 | 46.89 | 59.46 | 75.06 | 49.60 | 40.87 | 55.89 |
| TS-MOF(SHIKE+BS+LOS) | 74.55 | 59.71 | – | 69.95 | 76.22 | 51.39 | 42.11 | 59.90 | 79.03 | 51.77 | 34.77 | 56.21 |
| TS-MOF(SHIKE+BCL+LOS) | 74.67 | 59.45 | – | 69.95 | 76.15 | 50.73 | 42.28 | 59.63 | 79.17 | 49.66 | 34.73 | 55.51 |
| TS-MOF(SHIKE+CE-DRW+LOS) | 74.57 | 59.45 | – | 69.88 | 76.27 | 49.95 | 42.11 | 59.33 | 78.91 | 49.37 | 35.20 | 55.46 |
| TS-MOF(SHIKE+LDAM-DRW+LOS) | 74.99 | 59.35 | – | 70.14 | 76.15 | 51.15 | 42.22 | 59.79 | 78.40 | 50.74 | 34.73 | 55.62 |

(BCL+KPS), and our proposed fusion method TS-MOF. Compared with before fusion, the TS-MOF method shows a significantly increased separability of clusters in almost all categories in the feature space, indicating that our multi-objective fusion approach learns a more discriminative representation despite freezing the backbone in Stage 2. This suggests the classifier heads adapt in a way that better separates classes in the existing feature space.

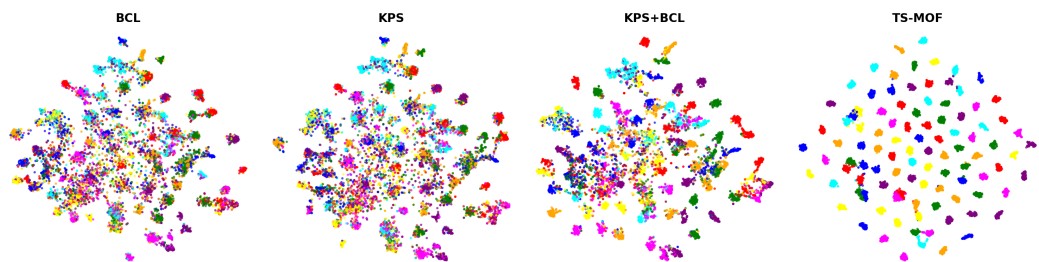

Figure 4: T-SNE comparative analysis of feature representations. We compare models trained on CIFAR-100-LT using BCL independently, KPS independently, a simple weighted fusion of BCL+KPS, and our proposed TS-MOF fusion method. Each point represents a feature vector from the test set, colored by its true class. Better separation indicates a more discriminative feature space for classification.

## 9.3 Confusion Matrix Analysis

Figure 5 shows a comparison of the predictive performance between individual strategies, simple fusion, and our TS-MOF method through confusion matrices. The confusion matrices are normalized by row (true label count) to show per-class accuracy patterns. From the figure, it can be seen that KPS pays more attention to some tail classes (higher diagonal values for later classes), while BCL focuses more on the head classes (higher diagonal values for earlier classes). Simple fusion may not effectively resolve conflicts and can potentially damage the performance of tail classes. Our method TS-MOF effectively combines the advantages of individual strategies and exhibits good performance across head, medium, and tail classes, as indicated by the higher and more uniform diagonal values compared to baselines and simple fusion.

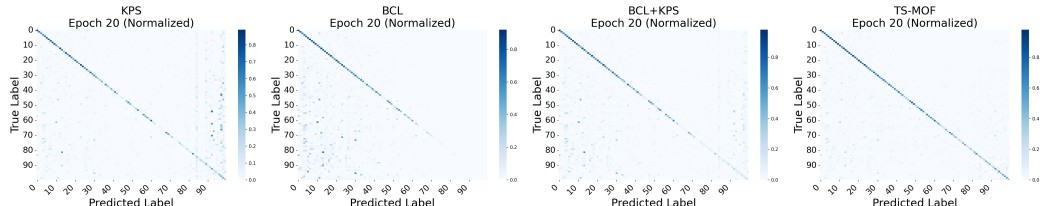

Figure 5: Comparative analysis of confusion matrices on CIFAR-100-LT (IR=100). We compare the predictive performance of models using BCL independently, KPS independently, a simple weighted fusion of BCL+KPS, and our proposed TS-MOF fusion method. Rows represent true labels, columns represent predicted labels. Diagonal elements indicate correct classifications (accuracy per class).

## 10 Limitations

Despite its advancements, TS-MOF has certain limitations. Its performance is inherently tied to the quality of the Stage 1 pre-trained features, as the encoder is frozen during fine-tuning. The framework also requires pre-selecting the specific set of LTR strategies to be included in the multi-objective optimization, which might require empirical tuning for optimal performance on new datasets. Furthermore, while designed for robustness, the framework still involves several hyperparameters that may need careful configuration.

## 11 Broader Impacts

TS-MOF aims to improve balanced recognition in long-tailed datasets, which are common in real-world applications. By enhancing tail class performance, our method can contribute to greater fairness and equity in ML systems by providing better representation for under-represented categories, potentially benefiting applications in healthcare, social sciences, and specialized domains. Making better use of rare data can also increase the utility of ML in resource-constrained settings. However, like any advanced recognition technology, there is a potential for misuse, such as in surveillance applications, emphasizing the need for responsible development and deployment.

