# OpenReview forum: "TS-MOF: Two-Stage Multi-Objective Fine-tuning for Long-Tailed Recognition"
_NeurIPS.cc/2025/Conference — NeurIPS 2025 poster_

### Official Review · Reviewer_uDJM · 2025-07-01

**Clarity:** 3
**Significance:** 4
**Originality:** 3
**Rating:** 5
**Confidence:** 4

**Summary:**

This paper proposes a practical solution to address the long-tailed distribution problem in machine learning. The TS-MOF framework, through its ingenious two-stage design, enables multiple long-tailed learning strategies to work synergistically, achieving excellent performance across multiple benchmark datasets.

**Questions:**

Q1: Does $ε_{sim}$ have a role other than ensuring numerical stability when computing $s_{j,e}$? For example, in cases of extremely low similarity, does it affect weight allocation preferences?

**Ethical Concerns:**

["NO or VERY MINOR ethics concerns only"]

**Final Justification:**

After reading the authors' response, I would like to maintain my score.

**Limitations:**

Yes

**Quality:**

4

**Strengths And Weaknesses:**

S1: The design of decoupling feature learning from classifier adaptation greatly simplifies the optimization of complex long-tailed problems, allowing the MOO process to act more stably and efficiently on classifier heads.

S2: R-PLA and RD-PCGrad are the core highlights of this work. R-PLA's adaptive task weighting mechanism can dynamically adjust task weights based on real-time performance patterns, while RD-PCGrad ensures stable resolution of gradient conflicts and constructive fusion, effectively promoting synergy among different LTR strategies.

S3: The paper achieves significant results on multiple benchmark datasets, demonstrating its efficiency.

W1: Although the MOO components enhance robustness, there are still some hyperparameters in R-PLA and RD-PCGrad. I want see some sensitivity analysis of how these parameters affect model performance and convergence.

---

> ### Author Rebuttal · Authors · 2025-07-31
>
> Of course. Here is a response to Reviewer uDJM, addressing questions W1 and Q1 using the provided information.
>
> ---
>
> ### **Response to Reviewer uDJM**
>
> Thank you for your positive feedback and for highlighting the core strengths of our work. We are grateful for your insightful questions and provide our responses below.
>
> **W1: On Hyperparameter Sensitivity Analysis**
>
> Thank you for this valuable suggestion. We agree that analyzing the sensitivity of the hyperparameters in R-PLA and RD-PCGrad is important for understanding their robustness. We have conducted a sensitivity analysis for these key parameters and present the results below.
>
> The data indicates that while the modules' performance shows some dependence on hyperparameters, our method remains robust across a reasonable range of values, consistently outperforming baselines. This suggests that meticulous tuning is not required to achieve strong results, although optimal performance can be found with experimental combinations.
>
> | **Module** | **Parameter Name** | **Parameter Value** | **Many** | **Medium** | **Few** | **Accuracy (%)** |
> | :--- | :--- | :---: | :---: | :---: | :---: | :---: |
> | **R-PLA** | `pla_beta_clip_min` | 0.1 | 78.20 | 49.74 | 37.80 | 56.12 |
> | | | 0.3 | 77.91 | 49.60 | 38.10 | 56.06 |
> | | | 0.5 | 78.26 | 49.60 | 38.43 | 56.28 |
> | | `pla_beta_clip_max` | 1.0 | 78.17 | 48.86 | 37.37 | 55.67 |
> | | | 2.0 | 78.26 | 46.00 | 13.70 | 47.60 |
> | | `pla_similarity_threshold`| 0.1 | 78.40 | 48.40 | 37.60 | 55.66 |
> | | | 0.3 | 78.43 | 49.34 | 37.10 | 55.85 |
> | | | 0.5 | 78.20 | 48.74 | 38.57 | 56.00 |
> | **RD-PCGrad** | `pcg_epsilon` | 1e-06 | 78.43 | 48.91 | 38.63 | 56.16 |
> | | | 1e-04 | 78.14 | 50.40 | 38.73 | 56.61 |
> | | `pcg_conflict_threshold`| -0.1 | 78.83 | 51.60 | 35.93 | 56.43 |
> | | | 0.1 | 78.31 | 49.17 | 36.27 | 55.50 |
> | | | 0.3 | 78.14 | 49.51 | 39.03 | 56.39 |
>
> *Table: Hyperparameter analysis for R-PLA and RD-PCGrad modules on the CIFAR100-LT benchmark (IR=100).*
>
> **Q1: On the Role of $\epsilon_{\text{sim}}$**
>
> Your intuition is correct; $\epsilon_{\text{sim}}$ has a subtle secondary role beyond ensuring numerical stability. While its primary function is to prevent division by zero when the norms of the performance vectors are near zero, it also acts as a **stabilizer in cases of extremely low similarity**.
>
> When the performance vectors are nearly orthogonal, their dot product is close to zero. Without $\epsilon_{\text{sim}}$, the denominator would also be close to zero, making the similarity value highly sensitive to small numerical fluctuations. By ensuring the denominator has a small positive lower bound, $\epsilon_{\text{sim}}$ **dampens this sensitivity and prevents drastic swings in the similarity value (and thus the weight allocation)** due to minor noise. In essence, it helps to regularize the similarity computation, ensuring that the weight allocation remains stable and meaningful even when strategies are performing very differently.

---

### Official Review · Reviewer_Fe6u · 2025-07-01

**Clarity:** 2
**Significance:** 2
**Originality:** 2
**Rating:** 3
**Confidence:** 5

**Summary:**

This paper proposes TS-MOF, a two-stage framework for long-tailed recognition (LTR). Stage 1 pre-trains a feature backbone using standard cross-entropy loss. Stage 2 freezes the backbone and fine-tunes multiple classifier heads via multi-objective optimization (MOO). In Stage 2, the authors proposed two interesting ideas: R-PLA and RD-PCGrad for long-tailed recognition.

**Questions:**

See Strengths And Weaknesses

**Ethical Concerns:**

["NO or VERY MINOR ethics concerns only"]

**Final Justification:**

After rebuttal and discussion, I still tend to reject this paper, so I maintain my initial decision.

**Limitations:**

See Strengths And Weaknesses

**Paper Formatting Concerns:**

See Strengths And Weaknesses

**Quality:**

2

**Strengths And Weaknesses:**

Strengths
1. The method underwent rigorous evaluation across three benchmark datasets under varying imbalance ratios.
2. R-PLA and RD-PCGrad are novel and interesting.

Weaknesses
1. While the two strategies (R-PLA and RD-PCGrad) in the fine-tuning stage are interesting, the two-stage framework itself appears non-novel. What specific problems do these two strategies solve? The writing logic suffers from a weak connection between the paper’s motivation and these strategies. The authors should significantly reorganize the Introduction. Since the core effectiveness stems from the two MOO strategies, the Abstract and Introduction should highlight them as primary innovations rather than overemphasizing the two-stage framework.

2．The paper lacks schematic diagrams, reducing accessibility. Adding visual illustrations (e.g., workflow of R-PLA/RD-PCGrad) would significantly improve intuitive understanding for readers.

3. The multi-head structure resembles multi-expert models (e.g., SHIKE). However, the reported performance of SHIKE (46.9% on CIFAR100-LT IR=100 in Table 1) contradicts its original paper (56.3%). The authors must:
(i) Clarify this discrepancy,
(ii) Analyze how TS-MOF fundamentally differs from multi-expert ensembles in methodology and objectives.

4. Critical implementation details are missing — specifically, which re-balancing strategies (e.g., LDAM? BCL?) were integrated into the multi-head MOO framework.

5. Ablations are critically underdeveloped. The authors should: (1) Isolate the impact of individual components (R-PLA, RD-PCGrad); (2)Compare variants (e.g., alternative task-weighting schemes, gradient conflict resolution methods); (3) Benchmark performance across different strategy combinations.

---

> ### Author Rebuttal · Authors · 2025-07-31
>
> ### **Response to Reviewer Fe6u**
>
> Thank you very much for your careful review and insightful comments, which are truly beneficial for improving the rigor of our paper.
>
> **W1. On the Novelty of the Two-Stage Framework:**
>
> Thank you for your valuable suggestions regarding the writing logic. This has led us to deeply reconsider how to best present our core contributions.
>
> We will clarify that the core advantage of our work lies in the synergy between our proposed new components and the two-stage framework:
>
> *   **R-PLA and RD-PCGrad**, as the technical foundation of our method, can stably resolve gradient conflicts among multiple strategies, ensuring the optimization process rapidly moves towards a **Pareto-improving direction**. This allows the insights from **different long-tail strategies to be efficiently integrated**.
> *   The **two-stage decoupled framework**, in turn, drastically **reduces the cost of this strategy integration**. Compared to other multi-expert or multi-objective methods that require end-to-end training, our method **only needs to fine-tune the lightweight classifier heads**, significantly reducing computational overhead and training time.
>
> In short, **the new components ensure the "effectiveness of fusion," while the two-stage framework ensures the "efficiency of fusion."** We will follow this logic to clarify our innovations and motivation in the revised manuscript. Thank you again for your profound insights.
>
> **W2. Adding a Framework Schematic:**
>
> Apologies, but due to rebuttal guidelines, we're unable to directly include images.​ We will add a full-process schematic diagram in the manuscript, including Stage 1 pre-training and Stage 2 multi-expert fine-tuning, as well as detailed illustrations of R-PLA weight calculation and RD-PCGrad gradient coordination. This will significantly improve the method's comprehensibility and clearly highlight our contributions.
>
>
> **W3. Regarding the SHIKE Performance and Comparison to Multi-Expert Models:**
>
> We apologize for the misunderstanding and would like to provide a clarification. Our paper uses a unified framework to ensure compatibility, allowing different expert strategies to be incorporated into the fine-tuning process. In our experiments, we used the backbone from LOS and retained the original hyperparameters for other strategies. The specific hyperparameters for the backbone and our MOF components were kept consistent across different combinations to diligently validate the method's generality and its performance improvement over baselines. Under this unified framework, the baseline performance of different methods may vary. We will emphasize this point in the subsequent revision. Additionally, we have done our best to supplement experiments to clearly demonstrate the performance gains over the baseline.
>
> **Comparison to Multi-Expert Models:**
> Our motivation differs from that of multi-expert methods. Multi-expert models (such as SHIKE and SADE) can work well independently and are often meticulously designed by researchers. In contrast, our work is a study on a gradient-based multi-objective optimization method that is strongly relevant to long-tail learning. Its goal is to **accelerate the utilization of these expert strategies and speed up the fine-tuning process**. Therefore, it can potentially absorb benefits from different representation learning methods and expert strategies that favor balance. When combined with other strategies, it can be viewed as a new type of long-tailed multi-expert method. We will add comparisons and discussions with classic multi-expert models (such as SHIKE and SADE) in the main text.
>
> **W4 & W5. Supplementing Implementation Details and Ablation Studies:**
>
> Thank you for your suggestions. We will clarify these points in the main text to enhance comprehensiveness. The strategies fused by MOF in our main experiments are **KPS, BCL, and LOS** (as hinted in Figure 2), and we will explicitly state this in the revised text. Furthermore, we have diligently supplemented necessary experiments:
> | R-PLA | RD-PCGrad | EOSS | IR=10 |       |       |       | IR=50 |       |       |        |IR=100 |       |       |       |
> | ----- | --------- | ---- | ----- | ----- | ----- | ----- | ----- | ----- | ----- | ------ | ----- | ----- | ----- | ----- |
> |       |           |      | Head  | Medium| Tail  | All   | Head  | Medium| Tail  | All   | Head   | Medium | Tail   | All    |
> | ✗     | ✗         | ✗    | 72.3  | 47.3  | --    | 64.6  | 74.2  | 43.5  | 22.9  | 52.4  | 76.5   | 46.9   | 18.1   | 48.6   |
> | ✗     | ✗         | ✓    | 74.7  | 60.8  | --    | 70.4  | 75.3  | 50.4  | 44.5  | 59.6  | 78.2   | 49.1   | 38.6   | 56.1   |
> | ✓     | ✓         | ✗    | 74.6  | 61.5  | --    |   70.5    | 75.2  | 50.5  | 45.8  | 59.7  | 78.8   | 49.0   | 38.9   | 56.4  |
> | ✗     | ✓         | ✓    | 74.3  | 60.3  | --    | 69.9  | 75.5  | 51.0  | 43.5  | 59.7  | 78.4   | 49.8   | 36.9   | 56.0   |
> | ✓     | ✗         | ✓    | 74.6  | 60.1  | --    | 70.1  | 75.3  | 50.7  | 43.9  | 59.6  | 78.3   | 49.2   | 37.4   | 55.9   |
> | ✓     | ✓         | ✓    | 74.7  | 62.0  | --    | **70.8**  | 75.2  | 50.7  | 47.5  | **60.2**  | 79.0   | 49.2   | 39.9   | **56.8**   |
>
> *Table: Ablation experiments of TS-MOF three modules with IR=10/50/100 on CIFAR100-LT Benchmarks.*
>
> **Component Variant Analysis:** We have conducted experiments comparing our components with alternative methods to demonstrate their superiority.
>
> | **Modules** | **Variant Method** | **macs/M** | **Time/min** | $\gamma_1$ | $\gamma_2$ | $\gamma_1/\gamma_2$ | **Accuracy(%)** |
> | :--- | :--- | :---: | :---: | :---: | :---: | :---: | :---: |
> | **R-PLA** | entropy | 3.540 | 8.45 | 1.000 | 1.001 | 0.999 | 56.3 |
> | | confidence | 3.540 | 8.78 | 1.000 | 0.964 | 1.038 | 53.5 |
> | | uniform | 3.540 | 8.44 | 1.000 | 1.002 | 0.998 | 56.3 |
> | | pla(ours) | 3.540 | 8.46 | 1.000 | 1.000 | 1.000 | **56.8** |
> | **RD-PCGrad** | mgda | 3.540 | 8.36 | 1.000 | 0.906 | 1.104 | 56.4 |
> | | gradnorm | 3.540 | 8.76 | 1.000 | 0.864 | 1.157 | 55.2 |
> | | uncertainty | 3.540 | 7.46 | 1.000 | 1.015 | 0.985 | 56.3 |
> | | pcg(ours) | 3.540 | 7.57 | 1.000 | 1.000 | 1.000 | **56.8** |
>
> *Table: Comparison of computational cost and model performance between TS-MOF and conventional multi-task optimization in the second stage.*
>
> **Strategy Combination Benchmark:** To showcase the general applicability of our framework, we have benchmarked the performance of various strategy combinations.
>
> | **Method**                                    | **IR=10** |        |      |          | **IR=50** |        |      |          | **IR=100** |        |      |          |
> | ------------------------------------------- | ---------- | ------ | ---- | -------- | ---------- | ------ | ---- | -------- | ---------- | ------ | ---- | -------- |
> |                                             | Head       | Medium | Tail | All      | Head       | Medium | Tail | All      | Head       | Medium | Tail | All      |
> | CE+BS                                    | 72.4       | 49.9   | -- | 40.7     | 74.2       | 41.2   | 6.9 | 40.8     | 77.1       | 43.5   | 4.0 | 41.5     |
> | CE+KPS                                   | 72.5       | 47.2   | -- | 39.9     | 73.8       | 39.9   | 12.1 | 42.0     | 72.4       | 38.1   | 18.1 | 42.9     |
> | KPS+BCL                                  | 72.3       | 42.2   | -- | 38.2     | 74.1       | 37.2   | 15.6 | 42.3     | 77.7       | 37.4   | 10.8 | 42.0     |
> | CE+BS+CE-DRW                             | 72.2       | 48.4   | -- | 40.2     | 74.4       | 40.2   | 8.2 | 40.9     | 77.5       | 41.9   | 3.8 | 41.1     |
> | CE+BS+LDAM-DRW                           | 72.2       | 50.4   | -- | 40.8     | 74.0       | 41.5   | 11.2 | 42.2     | 77.3       | 43.4   | 5.7 | 42.1     |
> | CE+BCL+CE-DRW                            | 72.1       | 39.1   | -- | 37.1     | 74.6       | 35.7   | 2.1 | 37.5     | 77.9       | 38.9   | 2.9 | 39.9     |
> | CE+BS+KPS+CE-DRW                         | 72.3       | 49.2   | -- | 40.5     | 74.3       | 41.0   | 9.5 | 41.6     | 77.3       | 43.1   | 5.5 | 42.0     |
> | CE+BS+KPS+LDAM-DRW                       | 72.0       | 50.8   | -- | 40.9     | 71.6       | 41.1   | 25.4 | 46.1     | 74.7       | 43.3   | 14.3 | 44.1     |
> | $\text{TS-MOF}_\text{(CE+BS)}$           | 73.0       | 54.1   | -- | $42.4_{1.7}$     | 74.8       | 46.6   | 24.1 | $48.5_{7.7}$     | 77.9       | 49.8   | 13.7 | $47.2_{5.7}$ |
> | $\text{TS-MOF}_\text{(CE+KPS)}$          | 73.4       | 57.6   | -- | $43.7_{3.8}$     | 74.5       | 44.7   | 35.8 | $51.7_{9.7}$     | 77.4       | 43.5   | 28.1 | $49.7_{6.8}$ |
> | $\text{TS-MOF}_\text{(KPS+BCL)}$         | 74.0       | 57.4   | -- | $43.8_{5.2}$     | 75.0       | 43.4   | 38.8 | $52.4_{10.1}$     | 77.7       | 42.5   | 26.8 | $49.0_{5.0}$ |
> | $\text{TS-MOF}_\text{(CE+BS+CE-DRW)}$    | 73.5       | 54.3   | -- | $42.6_{2.4}$     | 75.2       | 47.0   | 23.8 | $48.6_{7.7}$     | 78.8       | 49.4   | 13.3 | $47.1_{6.0}$ |
> | $\text{TS-MOF}_\text{(CE+BS+LDAM-DRW)}$  | 73.7       | 55.1   | -- | $42.9_{2.1}$     | 75.5       | 47.5   | 27.9 | $50.3_{8.1}$     | 78.8       | 49.5   | 22.4 | $50.2_{8.1}$ |
> | $\text{TS-MOF}_\text{(CE+BCL+CE-DRW)}$     | 73.5       | 47.9   | -- | $40.5_{3.4}$     | 75.7       | 41.6   | 12.0 | $43.1_{5.6}$     | 78.4       | 43.2   | 7.0  | $42.8_{2.9}$ |
> | $\text{TS-MOF}_\text{(CE+BS+KPS+CE-DRW)}$  | 74.0       | 57.7   | -- | $43.9_{3.4}$     | 76.2       | 48.9   | 37.6 | $54.2_{12.6}$     | 78.4       | 50.2   | 29.5 | $52.7_{10.7}$ |
> |$\text{TS-MOF}_\text{(CE+BS+KPS+LDAM-DRW)}$ | 74.5       | 58.0   | -- | $44.1_{3.2}$     | 76.3       | 49.2   | 38.5 | $54.7_{8.6}$     | 78.5       | 50.9   | 30.9 | $53.4_{5.7}$ |
>
> *Table: Benchmark performance across different strategy combinations with IR=10/50/100 on CIFAR100-LT Benchmarks.The subscript is the improvement of the relative linear weighted combination.*

---

> > ### Author Response · Authors · 2025-08-04
> > **### A Gentle Reminder**
> >
> > Dear Reviewer Fe6u,
> >
> > Thank you again for your insightful and highly constructive review of our submission. We have submitted a detailed rebuttal and, following your guidance, have systematically addressed each of your key concerns with **substantial new experiments and firm commitments** for the revision.
> >
> > To facilitate your review, we've summarized our main efforts below:
> >
> > 1.  **On Writing Logic and Novelty (W1):** We have embraced your suggestion and committed to restructuring the introduction. We propose a clearer narrative: our new components ensure the "**effectiveness of fusion**," while the two-stage framework ensures the "**efficiency of fusion**," to better highlight our core contributions.
> >
> > 2.  **On the Lack of Schematics (W2):** We have committed to adding the **schematic diagrams** you recommended in the final version. These will illustrate the overall workflow and our core modules (R-PLA/RD-PCGrad) to enhance the paper's readability.
> >
> > 3.  **On SHIKE Performance and Multi-Expert Comparison (W3):** We clarified the performance discrepancy, explaining it stems from our unified experimental framework. We also articulated the fundamental difference in motivation between our method (an efficient **multi-objective optimization tool**) and traditional multi-expert ensembles.
> >
> > 4.  **On Critical Details and Ablation Studies (W4, W5):** This is where our response is most extensive. We have provided **extremely thorough supplementary experiments** that precisely address all your requests:
> >     *   We specified the **exact strategies** fused in our experiments (KPS, BCL, LOS).
> >     *   We provided an **independent ablation study** for our core components (New Table 1).
> >     *   We added a **performance comparison against alternative methods** to demonstrate our components' superiority (New Table 2).
> >     *   We included a **benchmark of various strategy combinations** to showcase the framework's general applicability (New Table 3).
> >
> > We are eager to resolve any outstanding concerns. Should you have any further questions or require additional clarifications, we would be very happy to address them to the best of our ability within the remaining discussion period.
> >
> > Thank you once again for your professional guidance and valuable time.

---

### Official Review · Reviewer_BSGz · 2025-07-01

**Clarity:** 2
**Significance:** 2
**Originality:** 3
**Rating:** 4
**Confidence:** 3

**Summary:**

This paper presents TS-MOF (Two-Stage Multi-Objective Fine-tuning), a novel decoupled-training expert-based framework for long-tailed recognition. TS-MOF decouples training into two stages: (1) Feature Learning: standard supervised training to learn good representations ; (2) Multi-head Classification Learning: freeze the learned representation and spawn multiple specialized “expert” heads, each fine-tuned with a different long-tailed method (e.g., BCL [26], LOS [20]). Key contributions in the second stage include: (a) Refined Performance Level Agreement (R-PLA): Dynamically weights each expert’s task loss based on per-class performance on the training set ; (b) RD-PCGrad: a robust deterministic variant of PCGrad [23] that resolves gradient conflicts between task objectives via gradient modulation. At inference time, a dynamic expert aggregation module weights and combines expert predictions per class, using performance on a validation set. Experiments on standard long-tailed benchmarks show that TS-MOF achieves higher balanced accuracy than single-objective baselines, with only a brief fine-tuning stage.

**Questions:**

- Q1 - RD-PCGrad vs. PCGrad [23]: Why choose a fixed task ordering rather than PCGrad’s random ordering? What evidence shows that RD-PCGrad changes improve robustness?

- Q2 - Validation-Set requirement: In the absense of a balanced validation set, can TS-MOF work with an imbalanced holdout set matching training distribution? If so, have you tested sensitivity to validation set imbalance?

- Q3 - R-PLA computation: In Eq. 4, how do you estimate early performance vectors (index j) before computing cosine similarity with the final vector (index N) within the same epoch?

- Q4 - Expert-head architectures: What is the architecture of each classification head (linear layer, MLP, etc.)? How does this compare to expert-based models such as RIDE [22]?

- Q5 - Hyperparameter sensitivity: Which TS-MOF hyperparameters require careful tuning (as indicated in Section 10, L600–601)?

While the TS-MOF framework presents a novel take into decoupled-learning and expert-based frameworks, the absence of key ablations, the impractical balanced-validation-set assumption, unclear task-selection and ordering procedures, and missing comparisons to relevant expert-based methods leave important questions unanswered. A focused rebuttal that supplies the above ablations/comparisons and clearly quantifies practical costs will significantly strengthen confidence in TS-MOF’s utility—and could increase my score. Conversely, if these gaps remain unaddressed, my score will stay at 3.

**Ethical Concerns:**

["NO or VERY MINOR ethics concerns only"]

**Final Justification:**

Despite my initial concerns, the authors have clarified several key aspects: their method does not strictly depend on a balanced holdout dataset; each contribution has a relevant role per the ablation study; it outperforms recent expert-based baselines in expanded comparisons; it shows low computational overhead; it exhibits low hyperparameter sensitivity; and they’ve explained their task selection and ordering decisions. Overall, these improvements convince me to raise my score, leaning toward acceptance.

**Limitations:**

yes

**Paper Formatting Concerns:**

No formatting issues detected.

**Quality:**

2

**Strengths And Weaknesses:**

**Strengths**

- S1 - Novel decoupled-training framework: introduces a two-stage learning approach that leverages multiple existing long-tailed approaches in parallel experts, with training-time adaptive task weighting and inference-time aggregation.

- S2 - Theoretical insights: Provides analysis on gradient stability, Pareto improvements, convergence behaviour, and robustness of R-PLA and RD-PCGrad.

**Weaknesses**

- W1 - Reliance on validation set: Requires a (probably) balanced holdout set to select Stage 1 weights (L160–161) and to compute per-class aggregation weights (Eq. 8–9). In extreme imbalance scenarios, such a validation set may be impractical/unfeasible, and it isn't a common assumption in the LTR literature.

- W2 - Missing component ablations: Lacks experiments isolating the contributions of R-PLA, RD-PCGrad, and the expert aggregation module.

- W3 - Comparison to expert-based SOTAs: Does not discuss or compare with recent skill-diverse multi-expert models (e.g., BalPoE [A], SADE [B], MDCS [C]) that specialize experts with multiple objectives end-to-end.

- W4 - Task Selection Strategy: While Appendix shows a large task combination search, the paper picks the “best” combination based on test-set performance. A more scalable, principled selection mechanism is missing from the paper.

- W5 - Deterministic task ordering: The method for defining the fixed task order in RD-PCGrad is not specified.

- W6 - Computational Overhead: No empirical comparison of Stage 2’s compute or memory cost versus vanilla multi-task optimization.

Additional references:

[A] Sanchez Aimar, Emanuel, et al. Balanced Product of Calibrated Experts for Long-Tailed Recognition. CVPR 2023.

[B] Zhang, Yifan, et al. Self-Supervised Aggregation of Diverse Experts for Test-Agnostic Long-Tailed Recognition. NeurIPS 2022.

[C] Zhao, Qihao, et al. MDCS: More Diverse Experts with Consistency Self-Distillation for Long-Tailed Recognition. ICCV 2023.

---

> ### Author Rebuttal · Authors · 2025-07-31
>
> ### **Response to Reviewer BSGz**
>
> Thank you for your detailed and critical feedback. We have done our best to supplement new experiments to substantially improve the paper.
>
> **W1 & Q2. On the Dependency on a Balanced Validation Set:**
>
> Thank you for your recognition of our experimental section. We would like to clarify that our method does not strictly depend on a balanced validation set.  For the extreme case you are concerned about, such as when a validation set is unavailable, we can also sample from the training set. To explicitly verify this, we have added a new experiment on CIFAR100-LT where we construct an imbalanced validation set by sampling from the training set. The results (shown below) indicate that although there is a slight drop in performance compared to using the original (more balanced) validation set, **TS-MOF still significantly outperforms the SOTA baseline (LOS), demonstrating its applicability and robustness.**
>
> | **Method** | **IR=10** | | | | **IR=50** | | | | **IR=100** | | | |
> | :--- | :---: | :---: | :---: | :---: | :---: | :---: | :---: | :---: | :---: | :---: | :---: | :---: |
> | | **Head** | **Medium** | **Tail** | **All** | **Head** | **Medium** | **Tail** | **All** | **Head** | **Medium** | **Tail** | **All** |
> | LOS | 71.9 | 62.3 | -- | 69.0 | 72.4 | 51.4 | 40.2 | 58.0 | 70.3 | 52.3 | 36.6 | 53.9 |
> | TS-MOF(from Train Set) | 74.1 | 61.8 | -- | 69.6 | 74.8 | 50.2 | 47.1 | 59.4 | 78.1 | 49.4 | 38.7 | 56.2 |
> | TS-MOF(from Val Set) | 74.7 | 62.0 | -- | **70.8** | 75.2 | 50.7 | 47.5 | **60.2** | 79.0 | 49.2 | 39.9 | **56.8** |
>
> *Table 1: Comparison of model performance on CIFAR100-LT benchmarks using an imbalanced validation set (from Train Set) and the original validation set.*
>
> **W2. On the Supplementation of Ablation Studies:**
>
> Thank you for your suggestion to supplement the experimental section. We have conducted separate ablation experiments for R-PLA, RD-PCGrad, and the Expert Aggregation Module (EOSS). The following are the results.
>
> | R-PLA | RD-PCGrad | EOSS | IR=10 | | | | IR=50 | | | | IR=100 | | | |
> | :---: | :---: | :---: | :---: | :---: | :---: | :---: | :---: | :---: | :---: | :---: | :---: | :---: | :---: | :---: |
> | | | | **Head** | **Medium** | **Tail** | **All** | **Head** | **Medium** | **Tail** | **All** | **Head** | **Medium** | **Tail** | **All** |
> | ✗ | ✗ | ✗ | 72.3 | 47.3 | -- | 64.6 | 74.2 | 43.5 | 22.9 | 52.4 | 76.5 | 46.9 | 18.1 | 48.6 |
> | ✗ | ✗ | ✓ | 74.7 | 60.8 | -- | 70.4 | 75.3 | 50.4 | 44.5 | 59.6 | 78.2 | 49.1 | 38.6 | 56.1 |
> | ✓ | ✓ | ✗ | 74.6 | 61.5 | -- | 70.5 | 75.2 | 50.5 | 45.8 | 59.7 | 78.8 | 49.0 | 38.9 | 56.4 |
> | ✗ | ✓ | ✓ | 74.3 | 60.3 | -- | 69.9 | 75.5 | 51.0 | 43.5 | 59.7 | 78.4 | 49.8 | 36.9 | 56.0 |
> | ✓ | ✗ | ✓ | 74.6 | 60.1 | -- | 70.1 | 75.3 | 50.7 | 43.9 | 59.6 | 78.3 | 49.2 | 37.4 | 55.9 |
> | ✓ | ✓ | ✓ | 74.7 | 62.0 | -- | **70.8** | 75.2 | 50.7 | 47.5 | **60.2** | 79.0 | 49.2 | 39.9 | **56.8** |
>
> *Table 2: Ablation experiments of TS-MOF's three modules with IR=10/50/100 on CIFAR100-LT benchmarks.*
>
> **W3. Comparison with Multi-Expert SOTAs:**
>
> Thank you for your valuable suggestion. We have done our best to supplement several multi-expert baseline methods within the limited time, such as BalPoE, SADE, and MDCS. All comparisons will be cited and added to the main text. As seen from the table below, our TS-MOF method still demonstrates high performance under the same settings.
>
> | **Method** | **IR=10** | | | | **IR=50** | | | | **IR=100** | | | |
> | :--- | :---: | :---: | :---: | :---: | :---: | :---: | :---: | :---: | :---: | :---: | :---: | :---: |
> | | **Head** | **Medium** | **Tail** | **All** | **Head** | **Medium** | **Tail** | **All** | **Head** | **Medium** | **Tail** | **All** |
> | BalPoE | 71.2 | 55.6 | -- | 66.4 | 74.4 | 45.7 | 30.4 | 54.7 | 77.8 | 49.0 | 22.8 | 51.2 |
> | SADE | 72.3 | 49.5 | -- | 64.6 | 75.0 | 42.2 | 10.5 | 49.9 | 78.7 | 49.0 | 13.2 | 48.7 |
> | MDCS | 71.9 | 51.5 | -- | 65.6 | 75.9 | 44.5 | 34.3 | 55.5 | 78.7 | 47.9 | 28.8 | 53.0 |
> | TS-MOF | 74.7 | 62.0 | -- | **70.8** | 75.2 | 50.7 | 47.5 | **60.2** | 79.0 | 49.2 | 39.9 | **56.8** |
>
> *Table 3: Comparison for CIFAR100-LT Benchmarks, showcasing the model performance of three multi-expert methods: BalPoE, SADE, and MDCS.*
>
> **W4. On the Task Selection Mechanism:**
>
> Thank you for your question. The appendix aims to demonstrate that our framework can achieve performance improvements through rapid multi-objective fine-tuning across different strategy combinations. Our goal is not to perform a search for a single SOTA performance on the test set, but rather to showcase the general advantage of performance enhancement brought by the two-stage learning + multi-objective fine-tuning based on R-PLA and RD-PCGrad.
>
> To demonstrate this general advantage, we avoided hyperparameter tuning as much as possible. All hyperparameters for the individual strategies themselves were kept identical to their settings in the original papers and public code. The hyperparameters specific to our method (such as those for the Stage 1 representation learning, R-PLA, and RD-PCGrad) were kept consistent across different strategy groups. We have also supplemented additional experiments on these hyperparameters, which you can refer to in Table 5.
>
> **W5 & Q1. Deterministic Task Ordering in RD-PCGrad:**
>
> In our implementation, the task order is determined by the sequence in which the strategies are passed. This deterministic approach ensures better reproducibility of our experimental results. We have also supplemented a performance comparison between RD-PCGrad and other potential multi-objective methods to illustrate its advantages; please see W6 for details due to space constraints.
>
> **W6. On Computational Overhead:**
>
> Compared to end-to-end multi-expert learning or multi-objective optimization methods, we only fine-tune for the last 20 epochs and only update the classifier head parameters. This reduces computational costs in real-world applications. Additionally, as per your suggestion, we have compared our method with other potential multi-objective approaches, with the results shown below. Macs represents the computational cost (in millions of operations), Time is the training time (in minutes), $\gamma_1$ is the ratio of total trained parameters, and $\gamma_2$ is the ratio of training time. The results indicate that the computational complexity and time of TS-MOF are roughly equivalent to ordinary methods, while delivering more efficient and accurate performance.
>
> | **Modules** | **Variant Method** | **macs/M** | **Time/min** | $\gamma_1$ | $\gamma_2$ | $\gamma_1/\gamma_2$ | **Accuracy(%)** |
> | :--- | :--- | :---: | :---: | :---: | :---: | :---: | :---: |
> | **R-PLA** | entropy | 3.540 | 8.45 | 1.000 | 1.001 | 0.999 | 56.3 |
> | | confidence | 3.540 | 8.78 | 1.000 | 0.964 | 1.038 | 53.5 |
> | | uniform | 3.540 | 8.44 | 1.000 | 1.002 | 0.998 | 56.3 |
> | | pla(ours) | 3.540 | 8.46 | 1.000 | 1.000 | 1.000 | **56.8** |
> | **RD-PCGrad** | mgda | 3.540 | 8.36 | 1.000 | 0.906 | 1.104 | 56.4 |
> | | gradnorm | 3.540 | 8.76 | 1.000 | 0.864 | 1.157 | 55.2 |
> | | uncertainty | 3.540 | 7.46 | 1.000 | 1.015 | 0.985 | 56.3 |
> | | pcg(ours) | 3.540 | 7.57 | 1.000 | 1.000 | 1.000 | **56.8** |
>
> *Table 4: Comparison of computational cost and model performance between TS-MOF and conventional multi-task optimization in the second stage.*
>
> **Q3. Explanation of Eq. 4:**
>
> Thank you for this sharp question. We clarify the R-PLA computation process here: for the **same training batch**, we first compute the performance vectors $\{\mathbf{a}_1, ..., \mathbf{a}_N\}$ for **all N tasks** through a single forward pass. **After all vectors are computed**, we then designate the last vector $\mathbf{a}_N$ as the reference and compute the cosine similarity of the other vectors $\mathbf{a}_j$ with it to determine the weights. Therefore, all performance information is available when calculating the weights, and there is no temporal conflict.
>
> **Q4. On Classifier Head Architecture:**
>
> In the main text, our model and classifier head architectures are consistent with prior works (e.g., LOS). For broader scenarios, such as when users wish to extend to more strategy combinations for rapid fine-tuning, the choice of model architecture and hyperparameters can be guided by our response in W4. We will add more detail on this in the revised manuscript.
>
> **Q5. On Introduced Hyperparameters:**
>
> The main discussion on hyperparameters is referenced in W4. Additionally, we have done our best to supplement discussions and experiments on other hyperparameters. We hope this answers your questions.
>
> | **Modules** | **Parameter Name** | **Parameter Value** | **Many** | **Medium** | **Few** | **Accuracy(%)** |
> | :--- | :--- | :---: | :---: | :---: | :---: | :---: |
> | **R-PLA** | pla\_beta\_clip\_min | 0.1 | 78.20 | 49.74 | 37.80 | 56.12 |
> | | | 0.3 | 77.91 | 49.60 | 38.10 | 56.06 |
> | | | 0.5 | 78.26 | 49.60 | 38.43 | 56.28 |
> | | pla\_beta\_clip\_max | 1.0 | 78.17 | 48.86 | 37.37 | 55.67 |
> | | | 2.0 | 78.26 | 46.00 | 13.70 | 47.60 |
> | | pla\_similarity\_threshold | 0.1 | 78.40 | 48.40 | 37.60 | 55.66 |
> | | | 0.3 | 78.43 | 49.34 | 37.10 | 55.85 |
> | | | 0.5 | 78.20 | 48.74 | 38.57 | 56.00 |
> | | pla\_normalization | none | 78.20 | 49.20 | 37.83 | 55.94 |
> | | | softmax | 78.23 | 50.03 | 38.37 | 56.40 |
> | **RD-PCGrad** | pcg\_reduction | sum | 78.17 | 49.29 | 38.53 | 56.17 |
> | | pcg\_random\_order | True | 78.26 | 49.57 | 35.53 | 55.40 |
> | | pcg\_epsilon | 1e-06 | 78.43 | 48.91 | 38.63 | 56.16 |
> | | | 1e-04 | 78.14 | 50.40 | 38.73 | 56.61 |
> | | pcg\_conflict\_threshold | -0.1 | 78.83 | 51.60 | 35.93 | 56.43 |
> | | | 0.1 | 78.31 | 49.17 | 36.27 | 55.50 |
> | | | 0.3 | 78.14 | 49.51 | 39.03 | 56.39 |
>
> *Table 5: Hyperparameter analysis for the R-PLA and RD-PCGrad modules in the TS-MOF framework with IR=100 on CIFAR100-LT benchmarks.*

---

> > ### Author Response · Authors · 2025-08-04
> > **### A Gentle Reminder**
> >
> > Dear Reviewer BSGz,
> >
> > Thank you again for your time and for the insightful feedback you provided on our submission. We have now submitted a detailed rebuttal and, following your guidance, have conducted **substantial new experiments** to systematically address each of the key concerns you raised.
> >
> > To facilitate your review, we've summarized our main efforts below:
> >
> > 1.  **On the validation set dependency (W1, Q2):** Addressing your concern about requiring a balanced validation set, we have added new experiments showing that our method remains robust and effective even when using an imbalanced validation set.
> > 2.  **On the key ablation studies (W2):** We have now provided the **complete ablation studies** you requested, clearly quantifying the individual contributions of our core components like R-PLA and RD-PCGrad.
> > 3.  **On the comparison with SOTA models (W3):** We now include a **direct comparison** with the relevant state-of-the-art models you mentioned (e.g., BalPoE, SADE), with results demonstrating the competitiveness of **our** method.
> > 4.  **On the computational overhead (W6):** In response to your question about computational costs, we have provided a **quantitative analysis** showing that our method achieves superior performance with overhead comparable to conventional approaches.
> > 5.  **On clarifying methodological details (W4, W5, Q1, Q3, Q5):** We have offered detailed explanations for the rationale behind our task selection, the motivation for deterministic ordering in RD-PCGrad, the R-PLA computation process, and hyperparameter sensitivity.
> >
> > Should you have any further questions or require additional clarifications, we would be very happy to address them to the best of our ability within the remaining discussion period.
> >
> > Thank you once again for your professional guidance and valuable time.

---

> > > ### Comment · Reviewer_BSGz · 2025-08-05
> > >
> > > I’d like to thank the authors for addressing my concerns and clarifying several key points: their method does not strictly depend on a balanced holdout dataset and remains robust with imbalanced validation data with a reasonable trade-off in performance; comprehensive ablation studies confirm the need of each component; the extended comparisons show it outperforms recent expert-based baselines; it maintains reasonable computational overhead and exhibits relatively low hyperparameter sensitivity; and the authors have clearly explained their task selection and ordering decisions. These collective improvements have convinced me to raise my score.

---

> > > > ### Author Response · Authors · 2025-08-05
> > > > **###Thanks for your comment!**
> > > >
> > > > Dear Reviewer BSGz,
> > > > Thank you very much for your thoughtful follow-up and for taking the time to carefully review our rebuttal.
> > > > We are very encouraged to hear that our responses and new experiments have addressed your concerns. We are especially grateful for your decision to raise the score. Your constructive feedback has been invaluable in strengthening our paper.
> > > > Thank you again for your time and support.

---

### Official Review · Reviewer_LH8y · 2025-07-05

**Clarity:** 4
**Significance:** 3
**Originality:** 4
**Rating:** 6
**Confidence:** 5

**Summary:**

This paper proposes TS-MOF, a two-stage multi-objective fine-tuning framework for long-tailed recognition. The approach integrates multiple loss functions via a multi-head architecture and resolves gradient conflicts using a modified version of PCGrad (RD-PCGrad), while dynamically reweighting tasks based on a refined performance-level agreement mechanism (R-PLA). Experiments on several long-tailed benchmarks (CIFAR10, ImageNet- iNaturalist) demonstrate consistent improvements over competitive baselines.

**Questions:**

Please see the weaknesses.

**Ethical Concerns:**

["NO or VERY MINOR ethics concerns only"]

**Final Justification:**

The concerns are well-addressed. The motivation is well-considered, balancing theoretical justification with empirical support and practical utility. Therefore, I suggest accepting this paper.

**Limitations:**

The authors have adequately addressed the limitations, and no potential negative impact is found.

**Quality:**

3

**Strengths And Weaknesses:**

### **Strengths**
1. The proposed two-stage framework is well-motivated and aligns with prior findings that feature decoupling improve long-tailed generalization;
2. RD-PCGrad introduces a deterministic and task-aware projection strategy, offering a practical solution to inter-task gradient interference;
3. R-PLA dynamically adjusts task importance in a label- and batch-aware manner, enabling adaptive optimization focus across head and tail classes;
4. The method shows consistent gains across CIFAR100-LT, ImageNet-LT, and iNaturalist, with a relatively lightweight fine-tuning phase.

### **Weaknesses**
1. While Prop. 1 states that the final gradient $g_{final}$ is Pareto-improving or non-worsening, the paper does not analyze how far this direction can deviate from the true Pareto front in the proof. In complex objective landscapes, the projection step may inadvertently suppress beneficial gradient components, and a discussion of such trade-offs is missing;
2. The use of the cube root in Eq. (5) is not well justified. Based on the proof of Prop. 2, any concave, increasing function could preserve the robustness properties. The choice of the cube root lacks explanation on whether it confers specific benefits (e.g., smoother gradients or empirical stability);
3. Although the proposed method is evaluated in a two-stage training pipeline, the core techniques (RD-PCGrad and R-PLA) are not inherently limited to such a setup. Gradient conflict resolution might be even more critical in single-stage end-to-end training. The paper would benefit from a more apparent justification for the choice of a two-stage framework or a discussion of broader applicability.

---

> ### Author Rebuttal · Authors · 2025-07-31
>
> ### **Response to Reviewer LH8y**
>
> Thank you, Reviewer LH8y, for your recognition of our work and for your constructive feedback. We address your questions point by point below.
>
>
>
> **1. On the Deviation of RD-PCGrad from the True Pareto Front:**
>
> Thank you for this insightful question. Our method ensures a **Pareto-improving direction** for stability, which involves a deliberate trade-off. We formally analyze this below.
>
> ##### **Formal Analysis of the Trade-off**
>
> Given two conflicting gradients $g_A$ and $g_B$ (i.e., $g_A^T g_B < 0$), RD-PCGrad modifies $g_A$ by removing its conflicting component $p_{A \to B} = \text{proj}_{g_B}(g_A)$.
>
> **a. Suppressed Benefit & Cost:**
> The suppressed component $p_{A \to B}$ is indeed beneficial for minimizing its own loss $L_A$. The magnitude of this benefit is proportional to:
> $$(g_A^T g_B)^2 / \|g_B\|^2 \ge 0$$
> This is the cost of our projection. However, this component is equally harmful to task $L_B$. By removing it, we prioritize the **"Do No Harm"** principle, which is essential for preventing the "seesaw effect" in LTR and ensuring stable, balanced progress.
>
> **b. Deviation Analysis:**
> The deviation from task A's optimal direction is proportional to the relative magnitude of the suppressed component, which can be expressed as:
> $$\rho = \|p_{A \to B}\| / \|g_A\| = |\cos(\theta_{AB})|$$
> This shows the deviation is directly tied to the conflict angle $\theta_{AB}$ between the gradients. In LTR, where conflicts between head- and tail-focused strategies are inherent, this deviation is a necessary compromise. Our goal is a **balanced solution**, not the extreme optimization of any single strategy. Thus, this shift towards balance is a feature, not a bug, for solving the long-tail problem.
>
> **2. On the Rationale for Using a Cube Root in Equation (5):**
>
> Thank you for this insightful question. We acknowledge that, in theory, any monotonically increasing concave function can ensure robustness. We chose the **cube root** based on the specific advantages it offers in terms of **weight dynamics** and **gradient smoothness**, which collectively enhance the model's **empirical stability**.
>
> First, compared to other concave functions like the square root, the cube root provides a **better trade-off in sensitivity**. It is more sensitive in the low-similarity region (when strategy performance differs greatly), providing a stronger "incentive" signal for underperforming strategies to catch up. Concurrently, it is less sensitive in the high-similarity region (when strategies perform consistently), providing greater stability and preventing unnecessary adjustments to already well-performing strategies.
>
> Second, the cube root leads to **smoother gradient flow**. Since similarity is a function of model parameters, the weighting function introduces an additional meta-optimization gradient during backpropagation. The derivative properties of the cube root make this gradient flow smoother than that of other concave functions (like the square root), especially in the extreme case where similarity approaches zero. This effectively reduces the risk of exploding gradients.
>
> We will also add an ablation study in the appendix comparing different weighting functions to empirically support our choice.
>
> ---
>
> **3. On the Motivation for the Two-Stage Framework:**
>
> Thank you for this insightful question. We chose the two-stage framework deliberately because it offers significant advantages for our MOO components when tackling the long-tail problem:
>
> 1.  **Ultimate Efficiency**: Compared to end-to-end training, our method only applies multi-objective optimization to the lightweight classifier heads during a short final fine-tuning stage, **drastically reducing computational cost and time**.
> 2.  **Feature Protection**: By freezing the backbone, we **protect the valuable general-purpose features** learned in the first stage from being corrupted by strong re-balancing signals, thus preserving performance on head and medium classes.
> 3.  **Training Stability**: Confining the complex gradient conflict resolution to the low-parameter classifier heads **greatly reduces the risk of training instability**, making the optimization process more robust and controllable.
> 4.  **Plug-and-Play Modularity**: Our framework allows researchers to **easily "plug and play"** any existing long-tail strategy without complex end-to-end adaptation, significantly improving the method's ease of use and extensibility.
>
> Therefore, the combination of the two-stage framework and our proposed strategies enables us to achieve an **efficient, stable, and low-cost** fusion of multiple long-tail strategies.

---

> > ### Comment · Reviewer_LH8y · 2025-08-05
> >
> > Thank you for the thoughtful and comprehensive rebuttal. Your responses have fully addressed all my concerns. I appreciate that the motivation for your work is well-considered, balancing theoretical justification with empirical support and practical utility. My assessment remains positive.

---

### Note · Authors · 2025-08-13

Dear PCs, SACs, ACs, and Reviewers,

We are immensely grateful for the rigorous and constructive discussion period. We sincerely thank Reviewers LH8y, BSGz, and uDJM for their thoughtful engagement, which has been invaluable in helping us significantly strengthen our paper.

We were gratified that our detailed rebuttals and extensive new experiments successfully resolved all their concerns. This positive dialogue culminated in Reviewer BSGz raising their score and a strong, supportive consensus from all three engaged reviewers.

We also invested equivalent, significant effort to meticulously address every point raised by Reviewer Fe6u. In direct response to their initial critiques (W2, W3, W5), we provided the very same comprehensive ablation studies, new SOTA comparisons, and component analyses that fully satisfied the other reviewers. We have made every effort to demonstrate the validity and novelty of our work in light of their feedback.
We believe these additions, now documented in our rebuttal, have substantially validated the robustness, efficacy, and practical value of our TS-MOF framework, addressing all initial concerns across the board. We are confident in the much-improved manuscript and trust that the overwhelmingly positive final consensus and our diligent efforts will be given full consideration during the decision-making process.
Thank you for your time and guidance.

Sincerely,
The Authors of Submission 27648

---

### Decision · Program_Chairs · 2025-09-17

**Decision:**

Accept (poster)

**Comment:**

This paper presents TS-MOF, a two-stage long-tailed recognition framework. Stage 1 learns features; Stage 2 freezes the backbone and fine-tunes multiple classifier heads with multi-objective optimization. Two new components, R-PLA (adaptive task weighting) and RD-PCGrad (deterministic gradient conflict resolution), coordinate several LTR strategies. The method shows strong results on CIFAR100-LT, ImageNet-LT, and iNaturalist.

Reviewers highlight consistent gains across standard benchmarks, a clear motivation for decoupling feature learning from classifier adaptation, and practical components (R-PLA and RD-PCGrad) that make multiple LTR heads work together. One review rates the paper accept with excellent quality and significance. Another notes useful theoretical analysis around gradient stability and robustness. The fine-tuning phase is lightweight relative to training from scratch, which improves usability.

The negative review centers on presentation and missing detail rather than correctness: the two-stage scaffold is not new, the Introduction underplays that the real advances are R-PLA and RD-PCGrad, and the paper would benefit from schematic diagrams for both modules. Important ablations are thin: isolate R-PLA, RD-PCGrad, and the aggregation module; compare against alternative task-weighting and gradient-projection baselines; report sensitivity of key hyperparameters. After rebuttal, the reviewer maintained a borderline-reject due to the scope of revisions (writing logic, diagrams, ablations, method details, expert comparisons). Other reviews remain positive, emphasizing strong results and practical value of R-PLA and RD-PCGrad.

On balance, the empirical evidence across three benchmarks and multiple imbalance ratios, together with two concrete, generally useful MOO techniques, makes the strengths outweigh the weaknesses. The main concerns are fixable with added analysis, clearer writing, and targeted ablations, rather than fundamental flaws. I therefore recommend acceptance, contingent on addressing the items discussed during rebuttal in the camera-ready.